# A vascularized breast cancer spheroid platform for the ranked evaluation of tumor microenvironment-targeted drugs by light sheet fluorescence microscopy

David Ascheid[1,7], Magdalena Baumann[1,7], Jürgen Pinnecker[2], Mike Friedrich[2], Daniel Szi-Marton[1], Cornelia Medved[1], Maja Bundalo[3], Vanessa Ortmann[1], Asli Öztürk[1], Rajender Nandigama[1,4], Katherina Hemmen [2], Süleymann Ergün[1], Alma Zernecke [3], Matthias Hirth [5], Katrin G. Heinze [2] ✉ & Erik Henke [1,6] ✉

Targeting the supportive tumor microenvironment (TME) is an approach of high interest in cancer drug development. However, assessing TME-targeted drug candidates presents a unique set of challenges. We develop a comprehensive screening platform that allows monitoring, quantifying, and ranking drug-induced effects in self-organizing, vascularized tumor spheroids (VTSs). The confrontation of four human-derived cell populations makes it possible to recreate and study complex changes in TME composition and cell-cell interaction. The platform is modular and adaptable for tumor entity or genetic manipulation. Treatment effects are recorded by light sheet fluorescence microscopy and translated by an advanced image analysis routine in processable multi-parametric datasets. The system proved to be robust, with strong interassay reliability. We demonstrate the platform's utility for evaluating TME-targeted antifibrotic and antiangiogenic drugs side-by-side. The platform's output enabled the differential evaluation of even closely related drug candidates according to projected therapeutic needs.

In recent decades, the focus in cancer drug development has shifted from cytotoxic agents, which were relatively straightforward to evaluate, to candidates for targeted therapies that often cause more specific effects and can be more challenging to assess. In particular, agents targeting not tumor cells but the various cells of their supportive tumor microenvironment (TME) pose new challenges in creating suitable assay systems for in vitro testing, which remains a crucial initial phase in drug development.

Challenges in testing TME-targeted drugs are connected to the particularities of the TME and to the mode of action of these drugs. The TME consists of the tumor cells, blood vessel-forming endothelial cells, tumor-associated fibroblasts (TAFs), various infiltrating immune cells, and extracellular matrix (ECM). All these components interact in manifold forms and contribute to an often complex architecture that varies substantially from tumor to tumor[1]. Usually, TME-targeted drugs act by influencing these interactions within the TME: anti-angiogenic

[1]Institute of Anatomy and Cell Biology, Julius-Maximilians-Universität Würzburg, Würzburg, Germany. [2]Chair of Molecular Microscopy, Rudolf-Virchow-Center for Integrative and Translational Bioimaging, Julius-Maximilians-Universität Würzburg, Würzburg, Germany. [3]Institute of Experimental Biomedicine, Universitätsklinikum Würzburg, Würzburg, Germany. [4]Max Planck Institute of Heart and Lung Research, Bad Nauheim, Germany. [5]Institut für Medientechnik, Technische Universität Illmenau, Illmenau, Germany. [6]Graduate School for Life Sciences, Julius-Maximilians-Universität Würzburg, Würzburg, Germany. [7]These authors contributed equally: David Ascheid, Magdalena Baumann. ✉e-mail: katrin.heinze@uni-wuerzburg.de; erik.henke@uni-wuerzburg.de

drugs influence the sprouting or formation of blood vessels, anti-fibrotic drugs reduce the accumulation of TAFs or built-up of ECM, and immunotherapeutics increase the infiltration of immune cells and stimulate their attack on tumor cells[2–6]. Thus, testing TME-targeted drugs in monocultures of the primarily targeted cell population is not suitable as most of the affected interactions cannot be monitored in such a reduced system. Instead, an effective in vitro test system has to bring together various cell types of the TME and recreate, in part, the TME and its interactions.

There is another point for testing TME-targeted drugs in a system of confronting different cells: the various components of the TME constantly influence each other, leading to a balanced composition and architecture of the TME, although this balance is in a constant mode of remodeling and re-equilibrating[7,8]. Interference with one component by a TME-targeted drug also affects all other components until a new treatment-induced equilibrium is found[9,10]. These effects can be monitored only in an assay system combining various cell types. Moreover, the fact that all components of the TME are interdependent is a central argument for targeting non-malignant parts of the TME. It opens the possibility of strategically altering the tumor's characteristics to be less malignant or more sensitive to successive treatments.

Combining various cell types found in the TME is best achieved in a 3D setting. This permits not only to bring together different cell types but also offers the possibility to recreate the architecture of the TME. This is an important feature because TME-targeted drugs affect not only the abundance of TME components but also their spatial arrangements.

Most described 3D assay systems employ homogeneous tumor spheroids (MCTS) generated from isolated tumor cells grown without an adherable substrate. These MCTS allow for the formation of a rudimentary extracellular matrix (ECM) and the establishment of cell-cell contacts in all orientations, which significantly affects the response to therapeutic agents and thus allows for a more realistic assessment of drug effects on tumor cells[11]. However, the absence of other tumor-associated cells means that these models are unsuitable for the testing of TME-targeted drugs. An interest in more complex 3D-in-vitro models is documented in the growing number of publications describing various novel methods to recreate parts of the TME in 3D models[12–19]. The simpler of these models are restricted to the reproduction of individual elements of the TME, often a support matrix of fibroblasts, which still precludes their use in more comprehensive assays for TME-targeted drugs. The complexity of the few more advanced 3D-in-vitro tumor models (e.g. organ-on-a-chip systems), on the other hand, allows for even a moderate drug candidate throughput[20].

The fact that TME-targeted drugs are not designed to kill their target cells or stop their proliferation, but to influence the interactions within the TME in more subtle ways, creates an additional problem: a practical screening system must be capable of comprehensively capturing a wide range of possible drug-induced variations in the TME and in tumor cell behavior. Particularly suited to monitor these variations in the necessary detail is an imaging-based assessment of the model's state.

Increasingly, therapeutic questions that offer multiple ways of possible interference emerge. For example, fibrosis and desmoplasia in the TME can, in theory, be reduced by targeting any of a multitude of proteins or pathways[4,21]. That further multiplies the range of possible candidates, which should ideally be tested in direct competition. An ideal assay should then allow the ranking of these drugs aimed at different molecular targets and potentially different cell subsets within the TME, according to their efficacy, to achieve the therapeutic aim. Ranking drug candidates even in less diverse fields remains a challenge[22,23].

In conclusion, the realistic pre-clinical testing of TME-targeted drugs has to fulfill two requirements: a simple 3D model that replicates with high reproducibility parts of the TME by combining various tumor-associated cells and an analytic procedure that is able to record and decipher into processable parameters the changes under drug exposure.

Based on these considerations, we set several goals in developing a platform for screening TME-targeted drug candidates: (i) robustness (i.e., independence from the use of highly specialized reagents, procedures, and highly trained personnel), (ii) high intra- and interassay reproducibility, (iii) capacity for full automation, (iv) modularity, (v) versatility (i.e. the ability to test drugs targeting different molecular targets or cellular compartments), (vi) representation of different tumor subtypes, (vii) self-organizing reproduction of the TME, (viii) fully humanized setup, and (ix) capacity to rank drugs according to various effects of interest.

Here we present the development of a complete platform for the evaluation of TME-targeted drugs. The model for the TME was created in the form of 3D vascularized tumor spheroids (VTSs) that combined tumor cells, endothelial cells, fibroblasts, and macrophages, all of human origin. The VTSs display a complex architecture, including a pseudovasculature. VTSs are generated and exposed to drugs in a standard 96-well format. This makes the setup cost-efficient, robust, and permissive for simple handling of larger probe sets in a fully automated system. A parallel processing and immunofluorescence staining procedure enables the acquisition of high-resolution images of the various cellular compartments by light sheet fluorescence microscopy (LSFM). The LSFM output is translated by a comprehensive 3D-image analysis routine into multiparametric quantitative datasets that contain the full structural details of the VTSs. Finally, a routine for extracting and weighting critical descriptive parameters allowed for ranking candidates even in large testing fields. We demonstrate the platform's usefulness by evaluating and ranking the effect of various TME-targeted drugs reliably and with high reproducibility. Although the platform is designed for integration into an automated setup for the parallel assessment of a cohort of drug candidates in the biotech industry, the simple, self-organizing generation of the VTS models in standard labware and the routine evaluation by LSFM make it also interesting for in-depth evaluation of novel compounds or the role of TME-related genes in an academic setting.

## Results
### Generation of vascularized tumor spheroids for evaluation by LSFM
Many cancer cells do not form MCTS by themselves[24]. Co-culture with fibroblasts has the potential to ameliorate this problem[25]. We tested six established cell lines, of which five represented the various molecular subtypes of breast cancer, for their ability to form spheroids in a liquid overlay setup (Supplementary Table 1). Only one of these cell lines - MDA-MB-435s, a lymphoblastic melanoma cell line, previously misidentified as a Her2+ breast cancer cell line[26] - formed tumor spheroids by itself. The others failed to form stable agglomerates (Supplementary Fig. 1a). In contrast, the addition of normal human dermal fibroblasts (NHDF) resulted in the reliable formation of spheroids with incorporated tumor cells independent of the utilized tumor cell line. A tumor cells/fibroblast ratio of 1:20 was best suited for a dependable generation of MCTS (Supplementary Fig. 1b). The high surplus of NHDF accommodated the low proliferation rate of these cells compared to tumor cells.

Based on these findings, a screening platform centered on tumor spheroids was developed and entirely composed of human components (Fig. 1a). Five of the previously used cell lines were utilized as components of malignantly transformed cells, while NHDFs supplied stromal support and ECM components. To generate a pseudovascular (PV) network, human umbilical vein endothelial cells (HUVEC) were included in the cell suspension. Finally, we added THP-1 cells, a source for cells with characteristics of macrophages. THP-1s have been

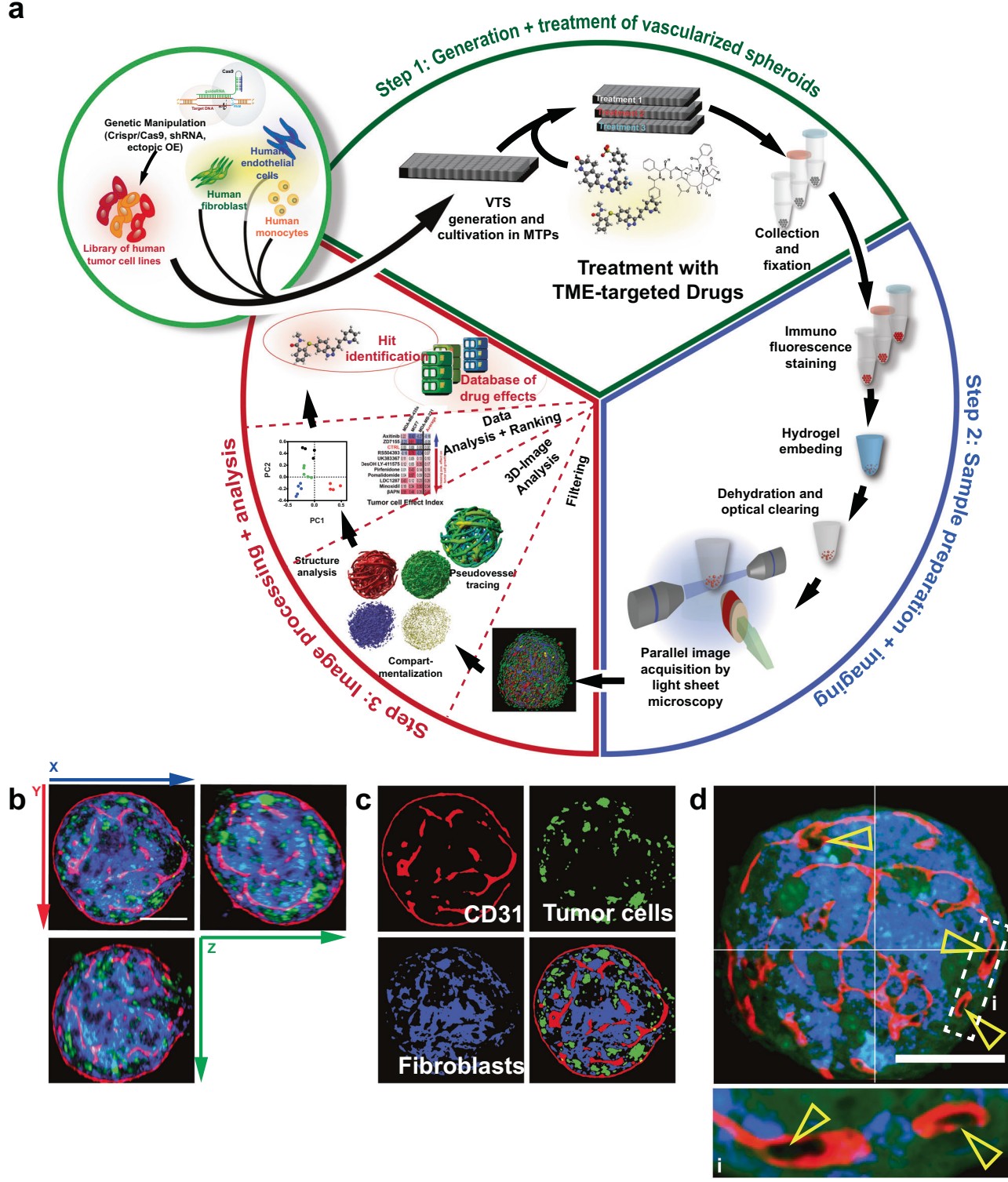

**Fig. 1 | Generation, treatment, preparation, and evaluation of vascularized tumor spheroids (VTSs). a** Schematic representation of the multistep process entailing pre-engineering of cell lines, generation of VTSs, treatment, sample preparation/staining, embedding, imaging, 3D-image processing, and image analysis, that enables the creation of datasets for a side-by-side comparison of complex structural effects triggered by TME-targeted drugs. **b** Preprocessed 3D-section image of a VTS generated with MDA-MB-435s cells (green, GFP-signal). Fibroblasts (blue, DsRed-signal). Incorporated ECs were stained for CD31 (red). **c** Compartmentalized representation of the xy-plane of the same VTS. The three cellular compartments are separated and processed. **d** Preprocessed section through a VTS generated with MDA-MB-468 cells. Yellow arrowheads show CD31+ PV structures with forming lumen. Scale bars = 100 μm.

previously used in 3D culture and can be polarized into both, an M1 or M2 phenotype[8,27,28]. Although having successfully utilized iPS-derived mesenchymal progenitor cells as a source for stromal and endothelial cells in spheroids previously[29], we decided on employing mature

primary cells for the drug screening platform. Cultivation of matured cells is less cost-intensive than iPS handling, especially if induction of pluripotency must be factored in. Moreover, prolonged maintenance, differentiation, and selection of the iPS-derived cells are complex and,

thereby, a source of variability in outcome. In contrast, the primary cells utilized in this project can easily be obtained commercially, even pre-screened for appropriate characteristics like response to cytokine stimulation. Therefore, differentiated primary cells promised better reproducibility and a better starting point for a streamlined, economical process. Simply combining the four different cellular components (tumor cells (TCs), NHDFs, HUVEC (ECs), and THP-1) and subsequent co-cultivation in agarose-coated multiwell dishes (MWDs) resulted reproducibly in the generation of complex tumor spheroids.

The central objective of this project was to establish a platform enabling the evaluation of drug effects on the TME. A microscopy-based assessment was best suited to follow changes in the various components of the VTSs in detail. Due to the size of the VTSs (~300–500 μm in diameter), we decided to use LSFM, which allows fast 3D imaging of large field of views while minimizing photobleaching and developed a versatile staining and embedding procedure that allowed secure handling and precise positioning and maneuvering of the samples within the LSFM imaging chamber. 10–15 VTSs were transferred in 1.5 mL conical microreaction vessels (MRVs) (Fig. 1a, Step 2). All subsequent steps on the pooled VTSs, including washing, fixation, immunostaining, and finally, hydrogel embedding, were performed in these MRVs. For optical clearing, a modified procedure based on ethyl cinnamate (ECi) was used[30]: Still, in the MRV, the VTS-containing hydrogel plugs were subjected to increasing concentrations of EtOH for dehydration. The dehydrated plugs were then cleared in two changes of ECi (Supplementary Table 2).

The ECi method proved in preliminary experiments to be superior to other organic solvent-based clearing techniques (based on benzyl alcohol/benzyl benzoate 1:2 (BABB)[31] or dibenzyl ether (3DISCO)[32]) as it left the hydrogel less brittle. Due to this brittleness, using BABB/3DISCO-cleared hydrogel plugs in pincer sample holders for LSFM proved challenging. This benefit was also observed in preparing optically cleared organ and tissue samples for LSFM. Moreover, the solvents used in ECi-based clearing are cheaper and less toxic than those used in the two other methods. Benefits in terms of optical properties in the cleared samples were not perceptible between ECi, BABB, or 3DISCO[33].

We first created VTSs by combining GFP-labeled tumor cells, DsRed-expressing NHDFs, HUVEC, and THP-1 cells (at 1000/20,000/2000/100 cells/well, respectively). The VTSs were collected, fixed, and stained for CD31 (PECAM, an endothelial cell marker) and CD11b (integrin $\alpha_M$ (ITGAM), macrophage-1 antigen (MAC-1)). Analysis of LSFM-acquired 3D images demonstrated that the VTSs generated by the newly established method displayed the anticipated complex architecture (Fig. 1b, c): after nine days of cultivation, spheroids had formed a PV network. A dense mesh of PVs was observable, not only covering the surface of the VTS but also expanding completely through the spheroidal structure (Supplementary Movie 1). The PV structures also clearly showed signs of lumen formation (Fig. 1d).

## Structures in VTSs undergo dynamic changes until a stable state is reached

After having established protocols suitable to generate complex VTSs and to visualize this complexity by IF-staining and LSFM, we developed a multistep image processing and evaluation routine (Fig. 2a). This routine encompassed identification and allocation of the various cell types included in the VTSs, segmentation of structure- or cluster-forming cells (e.g., pseudovessels), tracing of the PVs, and distance transformation to assess the distribution of PVs within the VTSs (Supplementary Fig. 1c–i, Supplementary Movies 2–4). Quantitative analysis of the identified and traced structures yielded a data set of ≥50 (depending on the number of cell types or molecular markers staining was performed for) parameters that numerically described the architecture of the VTSs in detail (Supplementary Table 3). To optimize the cultivation time of the VTSs with regard to

reproducibility and throughput, the dynamic changes occurring within the VTSs during cultivation were analyzed. Based on previous experiments, we hypothesized that the VTSs, after several days of cultivation, would reach a stable state in which the structure and composition of VTSs were fully established and change only marginally afterward. The time point when this stable state was reached would mark the timeframe for experiments because afterward, structural changes based on the dynamics of the cell interaction can be excluded and would no longer influence the outcome. VTSs were generated with the addition of either MCF7, MDA-MB-231, or MDA-MB-435s TCs. Samples of 15 VTSs were drawn after three, six, nine, and twelve days and processed (Fig. 2b, Supplementary Fig. 2a, b). Except for the MCF7-based VTSs, the size of the VTSs changed only slightly and was overall constant after six days of cultivation (Fig. 2b). However, examination by epifluorescence microscopy during cultivation showed that the interaction between labeled TCs and NHDF remained dynamic for a longer period (Supplementary Fig. 2a). After fixation and staining, the dynamics in the composition of the VTSs could be examined in more detail by LSFM. While the volume fraction of the CD31+ PVs in all models remained relatively constant over time, an equilibrium between fibroblasts and tumor cells was generally reached after nine days (Fig. 2c, Supplementary Movie 5). In particular, VTSs based on MDA-MB-435s cells showed a strong dynamic between the cellular compartments. Next, the changes in the complex PV architecture were analyzed over time. While the total volume of the PV network remained relatively constant, the structure of this network changed significantly during the cultivation period. Individual vessel sections grew larger in diameter over time, resulting in a broader distribution of vessel diameters in the PV networks (Fig. 2d, e, Supplementary Fig. 2c). Branching levels, i.e., the numerical indicator representing the individual vessel segment's rank within the hierarchy in its PV network, decreased over time, indicating a consolidation process based on pruning of small, low-rank vessels (Fig. 2f). Analysis of the distance of individual voxels within the VTS to the next PV surface showed that over time larger areas were farther separated from the PV, analogous to a reduced and increasingly heterogeneous vessel distribution in a real tumor (Fig. 2g). The process was strongly pronounced in MCF7- and MDA-MB-231-based VTSs, but less observable in MDA-MB-435s-based VTSs. Interestingly, this was an effect of PV structures retreating from the periphery of the VTSs, resulting in a VTS capsule distant from PVs (Supplementary Fig. 2d). In contrast to MCF7- and MDA-MB-231-based VTSs, PVs remained present at the surface of MDA-MB-435s VTSs. In total, changes in these parameters showed that the PV networks evolved during the cultivation time from a dense, uniform network of small PV segments to a more diverse pseudovasculature. These dynamic changes were most substantial in the MCF7 model and only moderately pronounced in the MDA-MB-435s-based VTSs.

Thus, the experiments demonstrated an intensely dynamic remodeling over the first nine days of cultivation, after which an equilibrated system had formed. Consequently, subsequent experiments were performed over a cultivation period of nine days.

## Tumor cell lines create their own specific TME

The previous experiments already demonstrated that including different tumor cell lines resulted in substantial variations in VTS composition and structure. We now investigated these differences in more detail by analyzing VTSs generated with any of the six tumor cell lines (Fig. 3a, Supplementary Fig. 3a). In addition, VTSs generated without tumor cells were examined to evaluate the general effect of malignant cells on the TME.

Depending on which tumor cells were added, size and structure of the VTSs differed strongly (Fig. 3a). The tumor cells determined the size VTSs reached after nine days of cultivation, either resulting in enlarged VTSs (MDA-MB-231) or significantly smaller VTSs (MDA-

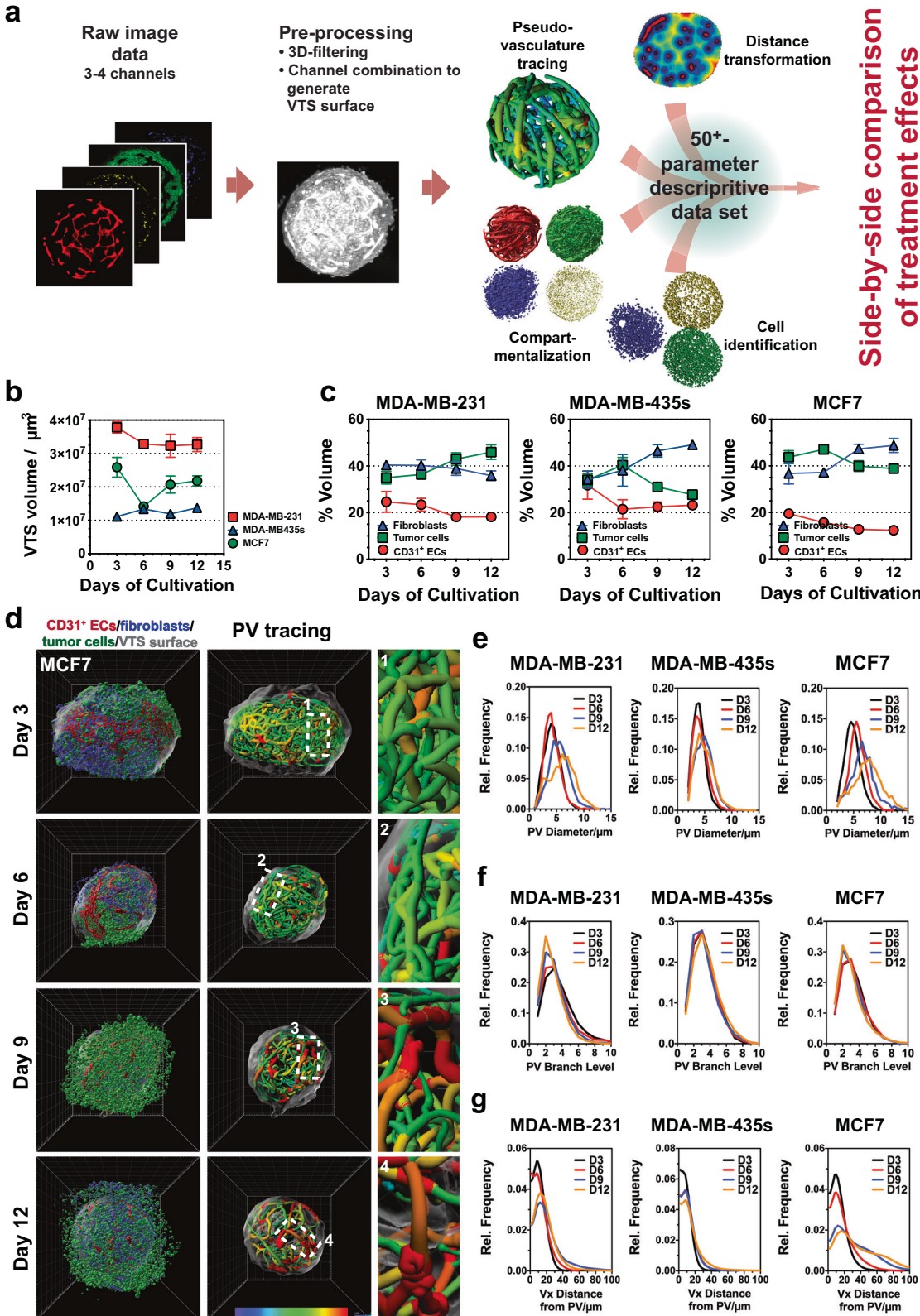

MB468, −435s) than spheroids generated only from NHDF (Fig. 3b). The VTSs also differed significantly in composition (Fig. 3c).

Interestingly, in VTSs generated with certain tumor cell lines (i.e., MDA-MB-435s and Sk-Br-3), the tumor cells formed isolated clusters within a matrix structure formed by the NHDF (Fig. 3d, e, Supplementary Fig. 3a). The other tumor cell lines formed VTSs, in which the tumor cells were more stochastically dispersed. Analysis of the median

distance separating tumor cells from fibroblasts and vice versa again demonstrated that in the MCF7, MDA-MB-231, −468, and ZR75-1 models, tumor cells and fibroblasts were interspersed regularly in small clusters, resulting in short distances between each other (Fig. 3e, Supplementary Fig. 3a, b). Conversely, in the MDA-MB-435s and Sk-Br-3 models, distances were larger and uneven, indicating the imbalanced amount of tumor cells versus fibroblasts. In MDA-MB-435s VTSs, PVs

**Fig. 2 | Progressive changes in composition and architecture of complex VTSs during cultivation. a** Image analysis routine: After pre-processing, 3D images underwent a four-pronged analysis routine of generating surface renderings of the (i) individual cellular compartments, (ii) localization of individual cells, (iii) tracing of the PV network, and (iv) evaluation of distal distribution from the pseudo-vasculature within the VTS. This resulted in a multiparametric data set for each VTS allowing for the side-by-side comparison of effects from cultivation conditions, VTS composition, or treatment. **b** Changes in VTS size over 12 days of cultivation. Volumes calculated from LSFM-acquired images in fixed VTSs. **c** Time-resolved changes of MCF7-based VTSs over a cultivation period of 12 days. 3D-rendering of surfaces of the three imaged cellular compartments and 3D representation of traced pseudovessels, colored according to segment mean diameter. **d** Changes in

VTS composition over 12 days of cultivation. Relative volume ratios of ECs, TCs, and fibroblasts over time. **e** Time-resolved distribution of PV-segment diameters over 12 days of cultivation in MDA-MB-231-, MDA-MB-435s- and MCF7-based VTSs. Displayed are the mean values of three VTSs/cell lines. **f** Time-resolved distribution of PV-segment branch levels over 12 days of cultivation in MDA-MB-231-, MDA-MB-435s- and MCF7-based VTSs. Branch levels give the hierarchical distance of a multi-segment PV branch from the longest (main-) branch of a PV network. Displayed are the mean values of three VTSs/cell lines. **g** Time-resolved distribution of voxel distances from the nearest PV over 12 days of cultivation in MDA-MB-231-, MDA-MB-435s-, and MCF7-based VTSs. Displayed are the mean values of three VTSs/cell lines. 3D grid spacing: 50 μm, error bars: ± SEM, $n = 3$ individual biological samples.

---

tended to be located within the NHDF matrix, avoiding direct contact with the tumor cells (Fig. 3d, Supplementary Movie 6). Consequently, the MDA-MB-435s tumor cells were located significantly farther away from PVs than tumor cells from the other lines (Fig. 3f, Supplementary Fig. 3b). The architecture of the PV network also differed in the various models (Fig. 3g, Supplementary Fig. 3c, d). The networks' structure ranged from chaotic, with a wide distribution of segment sizes (MDA-MB-231), to visibly organized but complex with a still strong hierarchy of vessel sizes (MCF7) and low-complexity networks with vessels of even size and diameter (ZR75-1). These discernable differences in complexity can also be quantitatively recorded, e.g. by the size of the formed networks, the diameter distribution of the individual vessel segments, or the hierarchical branching depth observed in them (Fig. 3g–j).

The number of THP-1 cells incorporated into the VTSs differed significantly depending on the tumor cell line used (Fig. 4a). Generally, including TCs reduced the amount of CD11b⁺ macrophages in the VTSs. MCF7 cells were the sole exception and displayed a high density of macrophages (Fig. 4b, c, Supplementary Movies 7 and 8). In tumors, macrophages are often located in proximity to blood vessels (and around necrotic areas, at the invasive front, and at the tumor/stroma border)[34]. Indeed, In the Sk-Br-3 and MDA-MB-435s models, macrophages showed a strong tendency to locate close to the PV (Fig. 4d). However, in the MCF7 model, macrophages appeared to be stochastically dispersed with respect to the distance to the PV networks. Vascular mimicry by CD11b⁺ cells, or incorporation into the PV, was not observed in any model.

As a component of the ECM distribution of the basal membrane-associated collagen IVa (Col IV) was studied. The amount of Col IV observable in the VTSs varied strongly by TC line used (Fig. 4e). Most of the Col IV appeared associated with the CD31⁺ PV, mimicking the vascular basal membrane association observed in tissue. However, amount of Col IV deposition and the degree of Col IV coverage of PVs in the VTSs varied substantially (Fig. 4f): in MCF7 (MDA-MB-468 and Sk-Br-3) VTSs, tight association over prolonged stretches of the PV segments was observable (Fig. 4g). In MDA-MB-231 and −435s VTSs only small spot-like patches of Col IV were displayed on the PVs. In MDA-MB-231 VTSs, substantial amounts of Col IV were deposited afar from the PV and surrounding mainly TCs (Fig. 4h). In several VTSs tubular protrusions of the Col IV sleeves appeared to extend the PVs (Fig. 4i, Supplementary Movie 9). These could be remnants of former, meanwhile regressed PVs, which might indicate previous remodeling and pruning of the pseudovasculature[35,36].

Another important aspect was the effect of the included macrophages on the VTSs architecture. MDA-MB-231-based VTSs were generated in the presence or absence of THP-1 cells. The inclusion of THP-1s in the VTS-generation process did not affect overall VTS size (Supplementary Fig. 4a) but reduced the packing density of tumor cells and fibroblasts (Supplementary Fig. 4b). Given the well-established effects of macrophages and monocytes on angiogenic processes, the consequence of THP-1-removal on the PV network was of interest. Indeed, the absence of THP-1s resulted in a nearly 40% reduction of relative

(and absolute) volume of CD31⁺-pseudovessels (11.31±0.53% vs. 18.57 ± 1.27%, Supplementary Fig. 4c). THP-1 cells seem to increase the proliferation and/or survival of the embedded endothelial cells. Without THP-1, fewer PV networks formed, whose complexity (branching degrees, overall network volume, branch diameter, etc.) was not significantly affected. Interestingly, vascular supply – as measured by distance to the nearest PV within the VTS - was also not reduced despite the diminished vessel density (Supplementary Fig. 4d). Closer inspection of the vessel and distance distribution within the VTSs revealed the reason for this seeming contradiction: when THP-1s were included in the VTSs, PV tended to be located closer to the surface of the VTS, leaving the center of the VTS undersupplied with pseudovessels (Supplementary Fig. 4e, f). In the absence of THP-1s, the PVs were no longer present on the VTSs' surface, but the PV density was increased in the center of the VTSs (Supplementary Fig. 4g, h).

## The screening platform shows high inter- and intra-assay reliability

In total, the characteristics of the individual VTSs were described in up to 71 parameters (Supplementary Table 3). Principle component analysis (PCA) of these descriptive parameters demonstrated high intra-assay reliability as all VTSs generated with supplementation of the same cell line clustered together (Fig. 4j). The MDA-MB-231 model shared similar characteristics with the VTSs generated under omission of tumor cells. Results of PCA showed that all other models differed significantly from each other, underscoring the individual characteristics that different tumor cell lines can induce in the TME.

The high intra-assay reliability demonstrated that the observed differences were indeed caused by the specific interaction of the various TC lines with their respective TME. However, an important criterion for developing an experimental setup into a useful drug screening tool is the degree of observed interassay reproducibility. To this point, we had acquired several fully independent but comparable datasets for the MCF7-, MDA-MB-231- and MDA-MB-435s-based VTS models. PCA of 46 parameters extracted by quantitative image analysis demonstrated a close clustering of the datasets acquired in different experiments with the same tumor cell line, while the different tumor cell lines were distinctively separated (Supplementary Fig. 5a). The standard deviation of the mean (SDm) between individual parameters obtained from independent experiments with the same tumor cell line was low to moderate, predominantly under 10% and significantly lower than the SDm between assays using different cell lines (Supplementary Fig. 5b). The SDm between assays with the same line was also in the same range as the intra-assay SDm.

## Hypoxia strongly influences VTS structure and size

After having established that the phenotype of tumor cells influences the TME and that these effects can reliably be picked up with our system, we started testing the effects of external factors and drugs in this system. First, we exposed MDA-MB-435s-based VTSs to hypoxia. After three days of cultivation, the VTSs were moved to an incubator

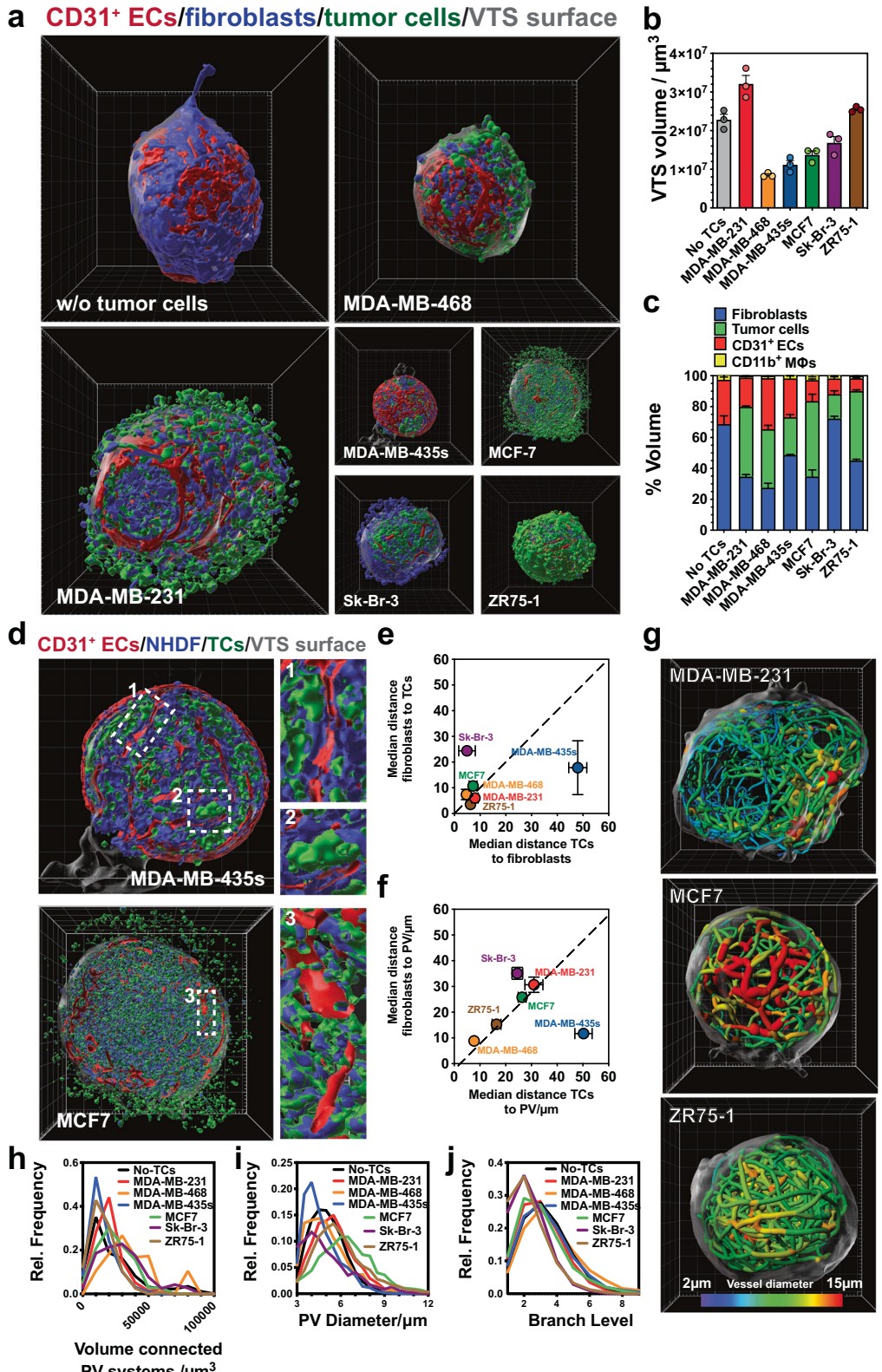

**a** CD31⁺ ECs/fibroblasts/tumor cells/VTS surface

**b**

**c**

**d** CD31⁺ ECs/NHDF/TCs/VTS surface

**e**

**f**

**g**

**h**

**i**

**j**

with a reduced oxygen level of 2% (Supplementary Fig 6a). The reduced oxygenation had a significant effect on the VTSs: after a total of nine days, the VTSs reached a volume nearly twice that of those under normoxia (20% $O_2$, Supplementary Fig 6b). The composition was shifted towards tumor cells at the expense of fibroblasts and endothelial cells (Supplementary Fig. 6c, Supplementary Movies 10 and 11). The PV network showed a phenotype of reduced complexity and decreased ability to supply the surrounding tissue (Supplementary Fig. 6d–g).

**TME-targeted drugs change VTS composition and architecture**

Two sets of TME-targeted drugs were tested: anti-angiogenic agents and inhibitors of collagen synthesis and stabilization. The tested anti-angiogenic drugs span a wide range with respect to both mechanism of

**Fig. 3 | Effect of different tumor cell lines on VTS composition and architecture.** **a.** Architecture of VTSs generated only from fibroblasts, ECs, and THP-1s (w/o tumor cells), or under the addition of five different tumor cell lines. 3D-rendering of surfaces of the three imaged cellular compartments. View on the outer surface of VTS. **b.** Volume of VTSs after nine days of cultivation. **c.** Relative volumes of cellular compartments within VTSs. **d.** Coronal cuts through the VTS center in the xy-plane with inlaid details. VTSs generated by the addition of MDA-MB-435s or MCF7 tumor cells. **e.** Comparison of distances of TCs and fibroblasts from nearest PV.

**f.** Comparison of distances of TCs and fibroblasts from the nearest complementary cell. **g.** Traced PV in VTSs generated with MDA-MB-231, MCF7, or ZR75-1 tumor cells. PV segments are color-coded according to mean diameter. **h.** Distribution of PV-networks total volumes within VTSs. Error bars: not shown for clarity. **i.** Distribution of PV-segment diameters within VTSs. Error bars: not shown for clarity. **j.** Distribution of PV-branch levels within VTSs. Error bars: not shown for clarity. 3D grid spacing: 50 μm, error bars: ± SEM, $n = 3$ individual biological samples. Source data are provided as a Source Data file.

action and state of clinical utilization: Axitinib (AXI), a potent VEGF-R2 inhibitor approved for the treatment of renal cell carcinoma[37], LDC 1267, also a tyrosine kinase inhibitor (TKI), targeting the TAM-receptor family (Tyro3, Axl, Mer)[38], and Deshydroxy (DesOH) LY-411575 a γ-secretase inhibitor that significantly affects tumor angiogenesis and pathological sprouting via the Dll4/Notch signaling pathway[39]. The three drugs were tested in both the MDA-MB-231 and −435s models.

The anti-angiogenic drugs were added after three days to the established VTSs ($c = 1 \mu M$), which were harvested six days later after a total cultivation time of nine days. All three agents significantly changed the cellular composition and distribution of the matured VTSs generated from MDA-MB-435s (Fig. 5a, b, Supplementary Movies 12–15) and MDA-MB-231 (Supplementary Fig. 7a, b) tumor cells. Overall viability was not affected, and VTS size was only in the case of AXI-treated MDA-MB-435s VTSs, and DesOH LY-411575 treated MDA-MB-231 VTSs reduced (Supplementary Fig. 7c). However, the composition was significantly moved towards a larger share of tumor cells and reduced abundance of fibroblasts (Supplementary Fig. 7d).

As expected, the potent, clinically approved angiogenesis-inhibitor AXI had a strong effect on the PV network within the VTSs: compared to untreated VTSs (Fig. 5a, Supplementary Fig. 7a), the amount of CD31$^+$ volume was strongly reduced (Fig. 5b, Supplementary Fig. 7a, b, d), indicative of the antiproliferative effect of AXI on ECs. Moreover, the CD31$^+$-ECs no longer formed a connected complex network, but the PV was broken up into smaller unconnected parts (Fig. 5a' vs. Fig. 5b' Fig. 5f, g and Supplementary Fig. 7b). Although the effect of LDC 1267 and DesOH LY-411575 on the PV was not as disruptive as that of AXI (Fig. 5c, Supplementary Fig. 7b), the size and complexity of the formed connected networks after treatment were in a similarly reduced range (Fig. 5d–g). The remarkable exception here was treatment with DesOH LY-411575, which reduced CD31$^+$ volume in MDA-MB-435s VTSs, but significantly increased the volume and complexity of the PV in MDA-MB-231 VTSs. In the MDA-MB-435s model, the treatment also resulted in a strong increase in PV diameters, which in the MDA-MB-231 model was barely affected by AXI and moderately decreased by LDC 1267 (Fig. 5h, i). Interestingly, although all drugs lowered the volume of CD31$^+$ endothelium in both VTS models, LDC 1267 treatment in the MDA-MB-231 model reduced the average distance to the endothelial surface significantly, indicating a more homogeneous distribution of the PVs within the VTS (Supplementary Fig. 7e). As a general trend, tumor cells were situated closer to the PV surface after treatment (Fig. 5j, k). This was most pronounced (and statistically significant) with AXI in the MDA-MB-435s model and with DesOH LY-411575 in the MDA-MB-231 model.

### Ranked screening of TME inhibitors reveals variations in response in different models

A second class of tested drugs were compounds affecting the ECM. VTSs were treated with agents targeting enzymes involved in collagen synthesis and stabilization: 2-aminopropionitrile (βAPN), a substance inhibiting all five members of the lysyloxidase family[10,40], minoxidil (2,6-Diamino-4-piperidinopyrimidin-1-oxid), an inhibitor of collagen lysyl hydroxylases (PLODs)[41], pirfenidone, an anti-fibrotic[42], pomalidomide, a TNFα-inhibitor that also displays anti-angiogenic properties (approved for the treatment of multiple myeloma and Kaposi sarcoma)[43,44], RS-504393, a CCR2-Inhibitor[45], ZD-7155, an Angiotensin-II

receptor-inhibitor[46], and UK 383367, a BMP-1 inhibitor[47] (Supplementary Table 4). VTSs based on MDA-MB-231, −435s, and MCF7 TCs were treated with these inhibitors ($c = 1 \mu M$, for 6d starting at d3 of cultivation), fixed, stained, imaged, and evaluated following the established protocol. PCA performed on the analysis results demonstrated that the effects of anti-fibrotic drugs, on one hand, and anti-angiogenic drugs, on the other, clustered together (Supplementary Fig. 8a). The separation by effect into different classes was most pronounced in MDA-MB-435s-based VTSs. The BMP-1 inhibitor UK 383367 had an effect spectrum distinct from the other tested substances. Although all tested drugs significantly affected the cellular composition in MDA-MB-231- and −435s-based VTSs, MCF7 VTSs seemed to be generally resilient with respect to changes in cellular composition (Supplementary Fig. 8b).

After having verified that our methods enabled us to generate and analyze VTSs with high reproducibility, we next intended to establish a method to rank the tested drugs according to their effects on PVs, TCs, macrophages, and the stromal component. The detailed parameter sets resulting from our analyses enable the comprehensive comparison of the effects of different agents on specific parameters describing the VTSs' architecture. On the other hand, the extensive, highly itemized list of parameters is not directly suitable for a ranking of agents according to a therapeutic intention. For ranking, two steps were necessary: (i) out of the full list of measured parameters, key parameters that strongly correlated with a therapeutic intention had to be identified, and (ii) weight factors for these key parameters had to be established, that made the calculation of a ranking value possible. To identify the key parameters and estimate weight factors, we took advantage of the ability of human observers to intuitively assess image data. (To illustrate this, AXI and LDC 1267 caused similar reductions of CD31$^+$ volume and size of PV networks, indicating similar anti-angiogenic efficacy (Fig. 5d–g). However, on observation of the 3D images, AXI had a much more disruptive effect on the PVs than LDC 1267 (Supplementary Movies 13 and 14). Thus, human observers can intuitively assess differences that are reflected but often not readily accessible in highly detailed datasets.) Because endothelial cells produced the most complex structures in the VTSs, we first tasked four untrained human observers with comparing the PV of treated VTSs with untreated counterparts and ranking the effects in a five-tier grading system (+1 (strongly improved), +0.5 (improved), 0 (unchanged), −0.5 (impaired), −1 (strongly impaired)). VTSs from MDA-MB-231 and MCF7 cells were used as training sets. From all comparisons of treated vs. untreated VTSs, a mean observer score (OS) was calculated (Fig. 6a). The OS was tested for correlation to all PV-descriptive parameters (Fig. 6b). Good correlations were calculated for the CD31$^+$-volume, the PV supply index, and the number of connected segments within the PV networks. Thus, observers seemed to take the overall size, homogeneity, and complexity of the PV networks into account when tasked with assessing an overall positive or negative effect. Using a system of non-linear equations, we stepwise calculated approximations for weight factors and correction factors for non-linearity (Fig. 6c). The resulting equation was used to calculate a vascular effect index (VEI) for the treatment groups in the MDA-MB-435s data set (=evaluation set). The calculated VEIs corresponded consistently with the assessment by observers (OS) (Fig. 6d).

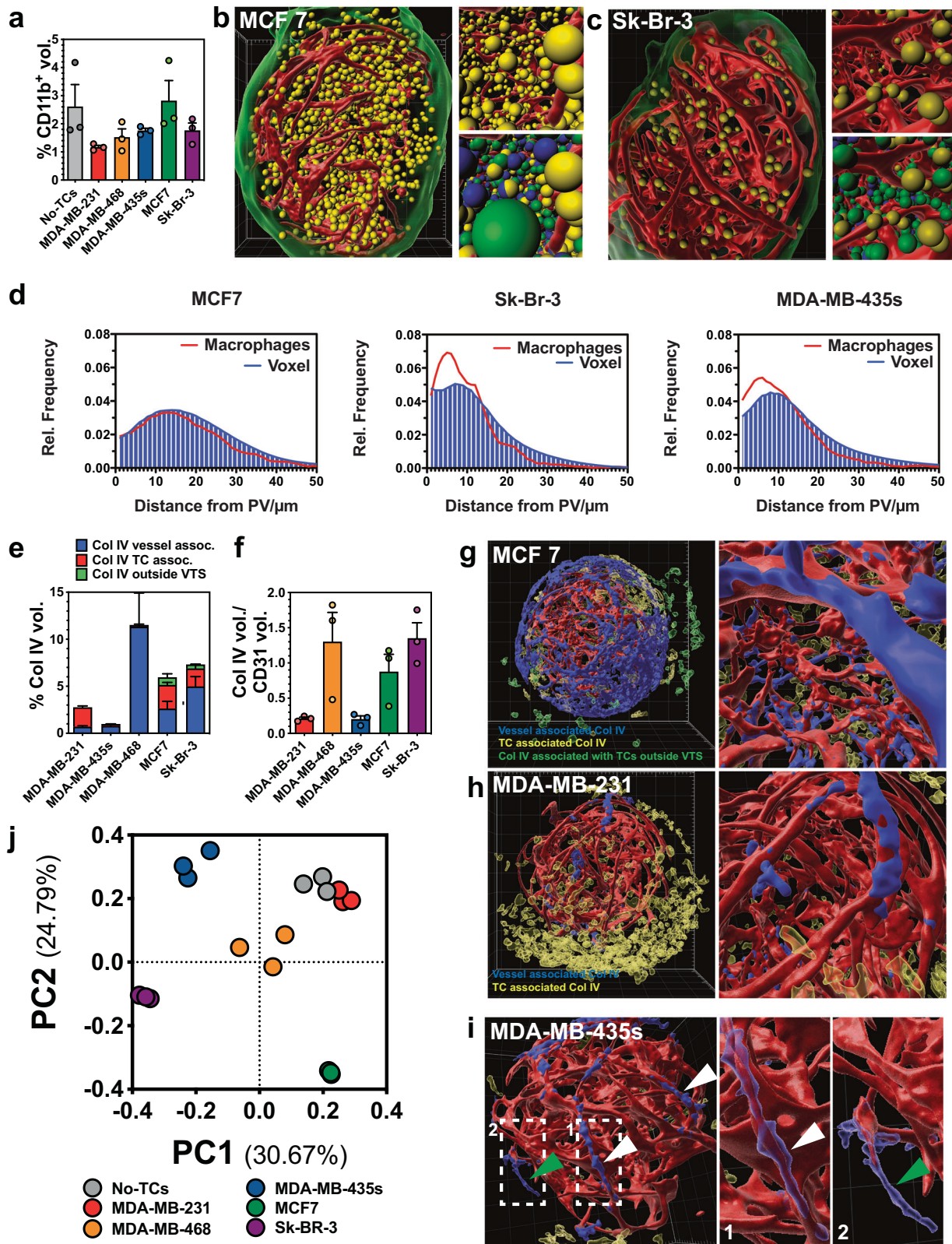

To substantiate screening results from our in vitro VTS platform and to validate the concept of ranking drug effects according to VEI, we used three of the tested drugs (AXI, UK 383367, and βAPN) in a murine breast adenocarcinoma model of orthotopically implanted AT3 tumor cells. Established tumors were treated with 3 rounds of the drugs (AXI at 25 mg/kg BW, UK 383367 at 25 mg/kg BW, and βAPN at 100 mg/kg BW, all applied q.o.d. by intraperitoneal injection). The perfused vasculature was stained intravitally by injection of an Alexa-Fluor647 labeled antibody to CD105 (Endoglin). After fixation and clearing, the resected tumors were examined by LSFM. All three drugs caused significant alterations to the network of perfused vessels in the examined tumors (Supplementary Fig. 8c, d). As expected, the reduction in perfused vessel density (PVD) was most pronounced after treatment with the approved anti-angiogenic drug AXI, although both

**Fig. 4 | Effect of different tumor cell lines on macrophage incorporation and ECM distribution. a** Relative CD11b⁺ volumes of VTSs generated only from fibroblasts, ECs, and THP-1s (w/o tumor cells), or under the addition of five different tumor cell lines. **b** Distribution of CD11b⁺ cells within a VTS generated with MCF7 cells. CD11b⁺ cells are shown as yellow orbs of 10 μm diameter, red: CD31⁺ PVs, green: VTS surface. Top insert: detailed view, bottom insert: detailed view with TCs (green orbs), fibroblasts (blue orbs). **c** Distribution of CD11b⁺ cells within a VTS generated with Sk-Br-3 cells. CD11b⁺ cells are shown as yellow orbs of 10 μm diameter, red: CD31⁺ PVs, green: VTS surface. Top insert: detailed view, bottom insert: detailed view with TCs (green orbs), fibroblasts (blue orbs). **d** Macrophage distance vs. overall voxel distances from the nearest PV within VTSs. Error bars: not shown for clarity. **e** Relative Col IV⁺ volume in VTSs generated under the addition of different tumor cell lines. **f** Fraction of Col IV⁺ volume directly associated with the PVs over CD31⁺ PV volume. **g** Distribution of Col IV in an MCF7-based VTS. Red: CD31⁺ PV; Blue: Col IV directly associated with the PV; transparent yellow: Col IV in VTS not associated with the PV, transparent green: Col IV produced by cells outside, but attached to the VTS. **h** Distribution of Col IV in an MDA-MB-231-based VTS. Red: CD31⁺ PV; Blue: Col IV directly associated with the PV; transparent yellow: Col IV in VTS not associated with the PV. **i** Distribution of Col IV in an MDA-MB-435s-based VTS. Red: CD31⁺ PV; Blue: Col IV directly associated with the PV; transparent yellow: Col IV in VTS not associated with the PV. White arrowheads: Col IV on the surface and surrounding PVs. Green arrowhead: empty Col IV sleeve. **j** Result from principal component analysis based on a 71-parameter descriptive data set. 3D grid spacing: 50 μm, error bars: ± SEM, *n* = 3 individual biological samples. Source data are provided as a Source Data file.

UK 383367 and βAPN also reduced PVD. After treatment with AXI or UK 383367 vessel appeared dilated (Supplementary Fig. 8d). Moreover, after AXI treatment, the vessel distribution in the normally densely vascularized AT3 tumors was heterogeneous, leaving large parts undersupplied (Supplementary Fig. 8e). Corresponding to the evaluation of the VTSs, the changes in the perfused tumor vasculature were quantified by compartmentalization, vessel tracing, and distance transformation within the tissue block. This allowed us to calculate also a VEI for the three drugs in the treatment of the AT3 tumors. The VEIs observed in the murine tumors corresponded with those observed in the VTSs (Supplementary Fig. 8f). In addition to the VEI, which is calculated from the changes in CD31⁺-volume, PV supply index, and complexity of the (pseudo)vascular networks, we also compared the changes in vessel segment diameters. Again, the data from the VTSs corresponded well with the effects observed in vivo – βAPN led to a small, non-significant increase in vessel diameter in the VTS models and a small, equally non-significant decrease in the murine tumor (Supplementary Fig. 8g). In conclusion, results from the murine treatment studies demonstrated the possibility of transferring results from our screening platform into an in vivo situation.

The VEI allowed for a ranked assessment of the tested drugs' effects in the different VTS models (Fig. 6e). We further wanted to distinguish between the anti-angiogenic effects of a drug and effects that more resemble vascular disruptive activities. Changes in relative CD31⁺ volume were used as a basis to calculate an index for anti-angiogenic behavior, and changes in the total length of the formed PV networks, reflecting their complexity, were used as a basis to calculate a vascular disruptive index (Fig. 6f). Both indices were scaled from −1 to +1. As expected, most drugs that showed anti-angiogenic behavior also caused a reduction in the complexity of the remaining PV networks and, therefore, ranked high in the vascular disruptive index, too. Similar indices were calculated for drug effects on tumor cells (from the density of tumor cells and their relative volume) and fibroblasts (from the density of fibroblasts and their relative volume) (Fig. 6g, h).

The compartmental effect indices provided a more condensed perspective. This allowed for a streamlined comparison of the sensitivity of the different VTS models. Interestingly, several drugs affected different VTSs in opposite directions. DesOH LY-411575 showed very different effects on the PV in MDA-MB-231 VTSs compared to the PV in MDA-MB-435s VTSs. This was evident in scoring by untrained observers (OS: +0.58 vs. −0.62) and in the calculated VEI (+0.90 vs. −0.59) (Fig. 6e). Thus, DesOH LY-411575 improved the PV in MDA-MB-231-based VTSs, but reduced both the volume of the PV as its complexity in MDA-MB-435s VTSs (Fig. 6f). In fact, closer inspection showed a denser, more ordered PV in MDA-MB-231 VTSs (Supplementary Fig. 8h, i). Furthermore, PV segments in the center of the MDA-MB-231 VTS looked more regular and more consistently connected to the outer PV layers (Supplementary Fig. 8h″, i″). In contrast, DesOH LY-411575 treatment of MDA-MB-435s VTSs resulted in a poorly defined PV with ECs forming large plate-like structures, barely resembling vessels, close to the VTSs surface (Supplementary Fig. 8j, k, k′). The PVs also

protruded less into the center of the MDA-MB-435s VTSs after treatment (Supplementary Fig. 8k″). Interestingly, in the MDA-MB-231 VTSs, a multitude of spur-like structures can be seen on the PV after treatment (Supplementary Fig. 8i′), which possibly are tip cells. Inhibition of Notch/Dll4 signaling with secretase inhibitors (DesOH LY-411575) leads to inflated but often unproductive angiogenesis in murine tumor models[48]. In MDA-MB-231 VTSs with their lower PV density, the results resemble an improved vascularization; in the already dense PV of the MDA-MB-435s VTSs, the increased angiogenic drive results in a PV that cannot differentiate anymore and looks strongly immature.

Context-dependent was also the vascular effect of the CCR2-inhibitor RS-504393: in the moderately vascularized MCF7 VTSs, the agent increased density and complexity of the PV network (Fig. 6i), resulting in a pro-angiogenic VEI of +0.62 (Fig. 6e). In the stronger vascularized MDA-MB-231 and −435s VTSs the results were moderate (Fig. 6j), with a VEI of +0.07 and +0.06 respectively. ZD-7155 (angiotensin-II-inhibitor) was, aside from AXI, the only tested drug that had a substantial inhibiting effect on MDA-MB-231 and −435s cells (tumor cell effect index of −0.33 and −0.23). Untreated MDA-MB-231 cells formed a densely interconnected network, evenly interspersed with fibroblasts (Fig. 6k, l). Treatment with ZD-7155 not only reduced the number of tumor cells (33.5 ± 9.6% vs. 50.9 ± 3.4% of total VTS volume in the MDA-MB-231 model, 21.7 ± 1.4% vs. 40.3 ± 0.9% in the MDA-MB-435s model). The TCs were no longer evenly distributed but formed small isolated clusters within the fibroblast matrix (Fig. 6l). In contrast ZD-7155 had a positive effect on MCF7 TCs (tumor cell effect index +0.61, Fig. 6g), increasing their density in the VTSs (Fig. 6m). The TAM-RTK-antagonist LDC 1267 was one of the tested agents that had a pro-proliferative effect also on MDA-MB-231 and −435s tumor cells, causing an increased absolute and relative TC volume (67.5 ± 1.2% vs. 48.4 ± 1.9% of total VTS volume in the MDA-MB-231 model, 66.3 ± 3.1% vs. 40.3 ± 0.9% in the MDA-MB-435s model), and an even denser network of TCs (Fig. 6k, l).

Thus, drug effect levels diverged significantly in different VTS models. Moreover, some drugs induced opposite effects. Therefore, we studied the effect of UK 383367 more comprehensively (including assessment of effects on Col IV and macrophages) by using a whole panel of five models, which indeed reacted differently to treatment (Fig. 7a). UK 383367 strongly reduced the density of CD11b⁺-macrophages in MCF7 and MDA-MB-468 VTSs but increased their abundance in the other models (MDA-MB-231, −435s, and Sk-Br-3). In MDA-MB-231 VTSs, the expansion of macrophages resulted in the forming of small macrophage aggregates (Fig. 7b). As expected, inhibiting BMP-1 with UK 383367 generally impeded proper collagen processing and reduced Col IV abundance in the VTSs, although the effect was small (and not statistically significant) in some models (Fig. 7c, d). Because previous results have shown that inhibiting collagen deposition can improve oxygenation in tumors[10], we investigated the stabilization of Hif1α in MDA-MB-435s VTSs. UK 383367 treatment significantly reduced the abundance of Hif1α⁺-cells (Fig. 7e, f, Supplementary Movies 16 and 17).

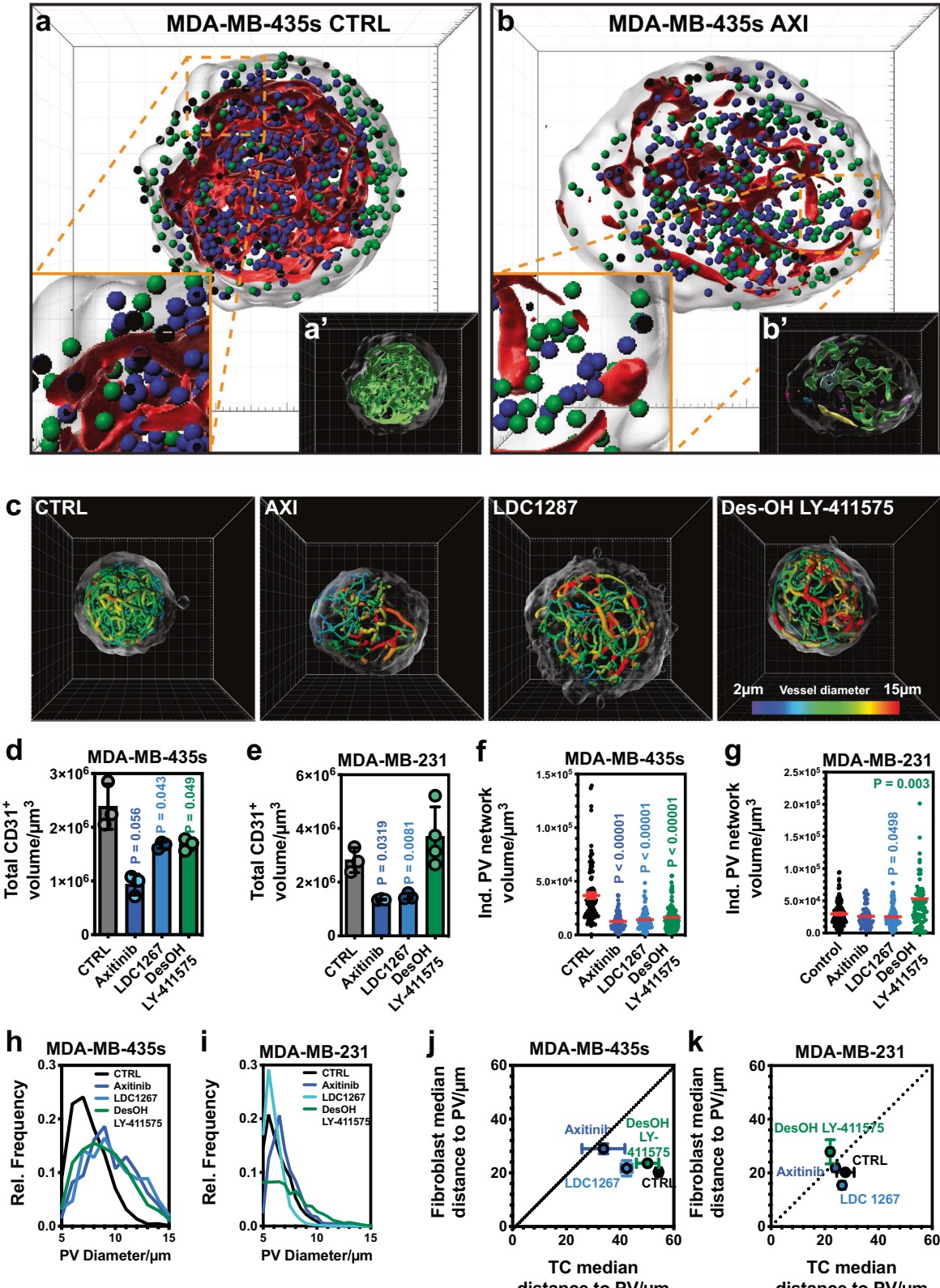

## Response to cytotoxic drugs is dependent on the TME

The TME also strongly influences the response of tumor cells to treatment, e.g., exposure to cytotoxic drugs. We slightly modified our assay system to specifically and quantitatively evaluate toxicity on the tumor cells embedded in the VTSs: using luciferase-expressing tumor cells to form the VTSs, it was possible to estimate TC viability by correlating it to the emitted luminescence signal and to distinguish effects on TCs from those on the other cell types (Supplementary Fig. 9a, b).

MDA-MB-435s-based VTSs were treated with paclitaxel (PTX), while MDA-MB-231-based VTSs were treated with cisplatin (CDDP). Compared to 2D cultures of the respective cell lines, $EC_{50}$-values for

**Fig. 5 | Effect of anti-angiogenic agents on VTS composition and architecture.**
**a** Architecture of untreated MDA-MB-435s-based VTSs. 3D-rendering of CD31⁺-PV surfaces (red) with TCs (green orbs), and fibroblasts (blue orbs). Frontal cut through VTS center in the xy-plane. **a′** 3D-rendering of CD31⁺-PV surfaces in the same view, colored to represent the different structural objects. **b** Architecture of MDA-MB-435s-based VTSs after treatment with AXI. 3D-rendering of CD31⁺-PV surfaces (red) with TCs (green orbs), and fibroblasts (blue orbs). Frontal cut through VTS center in the xy-plane. **b′** 3D-rendering of CD31⁺-PV surfaces in the same view, colored to represent the different structural objects. **c** Tracing of PV networks in MDA-MB-435s-based VTSs treated with three different anti-angiogenic agents. 3D-rendering of surfaces and segments colored according to diameter. **d** Absolute volumes of CD31⁺-cells within MDA-MB-435s-based VTSs after treatment. **e** Absolute volumes of CD31⁺-cells within MDA-MB-231-based VTSs after

treatment. **f** Volumes of individual CD31⁺-PV networks identified by tracing in MDA-MB-435s-based VTSs after treatment. **g** Volumes of individual CD31⁺-PV networks identified by tracing in MDA-MB-231-based VTSs after treatment. **h** Distribution of PV-segment diameters in MDA-MB-435s-based VTSs after treatment. $N = 3$, error bars: not shown for clarity. **i** Distribution of PV-segment diameters in MDA-MB-231-based VTSs after treatment. Error bars: not shown for clarity. **j** Comparison of distances of TCs and fibroblasts in MDA-MB-435s-based VTSs from nearest PV after treatment. **k** Comparison of distances of TCs and fibroblasts in MDA-MB-231-based VTSs from nearest PV after treatment. 3D grid spacing: 50 μm, error bars: ± SEM, analyzed with unpaired two-tailed *t*-test against CTRL, $n = 3$ (except MDA-MB-231 DesOH LY-411575: $n = 4$) individual biological samples. Source data are provided as a Source Data file.

the drugs increased by approximately two orders of magnitude (Fig. 7g, h). With MDA-MB-435s cells that formed MCTS without the addition of stromal cells, it was also possible to test if this was primarily a diffusion effect. MCTS formed only from MDA-MB-435s cells, were indeed more sensitive to PTX than VTSs formed from MDA-MB-435s and NHDF ($EC_{50}$: 1.08 ± 0.17 μM vs. 3.46 ± 0.79 μM, $P = 0.03$). Thus, the low sensitivity seen in complex VTSs is probably not only an effect of reduced diffusion but also one of the protective cues from the modeled TME. FACS-analysis showed that cultivation as VTSs in 3D significantly increased the expression of CD44 (a stem marker linked to chemoresistance[49]) in the GFP⁺-tumor cell population within the VTSs compared to tumor cells cultivated in 2D (Fig. 7i). In contrast to the other cell lines CD44 expression was already high in MDA-MB-435s cells cultivated in 2D, and their abundance did not increase in 3D MCTS (formed only from MDA-MB-435s cells) or in co-cultivation as VTSs. More detailed analysis showed that the ALDH1A1⁺ (another marker linked to chemoresistance) subpopulation strongly increased upon co-cultivation with NHDF in 3D (Fig. 7j, Supplementary Fig. 9c, d). ALDH1A1-expression was not increased in MDA-MB-435s-only MCTS, demonstrating that this was not an effect of the 3D culture conditions; the proximity of NHDF was necessary. The addition of HUVEC and/or THP-1 to the MDA-MB-435s/NHDF spheroids again reduced ALDH1A1-expression.

### Targeted genetic manipulation in various TME components

The growth factors (GFs) PDGF-B and VEGF-A₁₆₅ were overexpressed in ZR75-1 cells, which were then used to generate VTSs. As expected, VEGF-A overexpression (OE), but also, to a lesser degree, PDGF-B increased the density of the CD31⁺-PV (Fig. 8a, b). ZR75-1 VTSs are characterized by a well-structured PV network of segments with a homogeneous diameter distribution (Figs. 3g and 8c). OE of VEGF-A changed this to a more diversified PV network with increased average vessel diameters. This finding is in line with the effect of VEGF-A in tumors. OE of any of the two GFs also increased the density of CD11b⁺-macrophages (Fig. 8d, e). Moreover, the GFs also attracted macrophages to migrate deeper from the surface toward the center of the VTS (Fig. 8d, f).

The transcription factors Snai1 and Twist1 are well-characterized key regulators of epithelial-mesenchymal transition (EMT) in various cancers[50,51]. In tumor cells, they induce or repress complex cellular programs that result in phenotypical and morphological changes and in the expression of factors controlling the tumor cell-TME interaction. This includes factors involved in angiogenesis[52,53]. Because of these manifold effects on both the tumor cells and the TME, we used overexpression of Snai1 and Twist1 to test the potential of our platform to evaluate the effect of the misregulation of individual genes in different cellular components of the tumor. Moreover, as the effects of Snai1/Twist1 overexpression were expected to be mild, these studies would allow us to judge the sensitivity of the established evaluation process.

To study the differences of overexpressing the transcription factors in either the tumor cells or the stromal components, stable DsRed⁺

or DsRed⁺/Snai1⁺ or DsRed⁺/Twist1⁺ MDA-MB-231 cells and NHDFs were generated by transduction with lentiviral particles (Fig. 8g, Supplementary Fig. 10a, b). The resulting red-fluorescent MDA-MB-231 cells were mixed with GFP⁺-NHDF, HUVEC, and THP-1, and the red-fluorescent NHDF with GFP⁺-MDA-MB-231, HUVEC, and THP-1 to generate six different lines of VTSs. The experimental setup necessitated the use of two control groups (231-GFP/NHDF-DsRed and 231-DsRed/NHDF-GFP).

VTSs were of similar size, with the sole exception of overexpression of Snai1 in NHDF, driving a significant reduction of VTS volume (Supplementary Fig. 10c). PCA of the fully parametrized data set demonstrated that OE of these two transcription factors has indeed significant and, importantly, distinct effects on the developing VTSs. These effects also differed when overexpressed in different compartments (TCs vs. fibroblasts, Fig. 8h). In addition, PCA showed that VTSs generated from 231-GFP/NHDF-DsRed or 231-DsRed/NHDF-GFP had a very similar architecture, as the six samples from the two groups clustered together. Individual descriptive parameters were highly similar between the two control groups (Supplementary Fig 10d).

A closer examination of the PCA results revealed that differences in parameters describing the PV contributed strongly to the differences between groups (Supplementary Fig 10e, f). In contrast to Twist1 OE, OE of Snai1 in the tumor cells increased the relative and absolute volume of CD31⁺ PVs in the VTSs. This was observed for both OE in NHDF and MDA-MB-231 TCs (Supplementary Fig 10g). Snai1 has been shown to positively regulate VEGF-A, while Twist1 might not have a direct effect on expression levels of angiogenic factors like VEGF-A or FGF2[54,55].

Independent of the cell type of overexpression, Twist1 induced a more homogeneous PV distribution within the vessels, while Snai1 overexpression tended to increase heterogeneity (Fig. 8k, Supplementary Fig 10h). Despite this enhanced heterogeneity, the overall increased CD31⁺ PV volume after Snai1 OE in TCs resulted in TCs being located closer to PVs (Fig. 8l). Twist1 OE, on the other side, affected the distribution of the PV network within the VTSs. In control VTSs, a significant part of the PV was directly at the surface or in the outer regions of the VTS, leaving the center poorly supplied (Fig. 8m, n). Overexpression of Twist1 in the fibroblast compartment resulted in the PV being more centralized, without larger regions located far away from the next PV branch (Fig. 8o, p).

## Discussion

In this project, we describe the development of a VTS-based assay platform for the detailed assessment of TME-targeting drugs. The modular platform allows for various modifications in VTS generation, sample processing, and image-based analysis. This flexibility enables the adaptation of the platform to a wide range of screening tasks. Moreover, the newly designed analytic workflow by itself adds to this flexibility in several ways, as it yields a multiparametric description of the structure of the VTS, its cellular components, and their relationship to each other. First, this multiparametric disassembly enables a

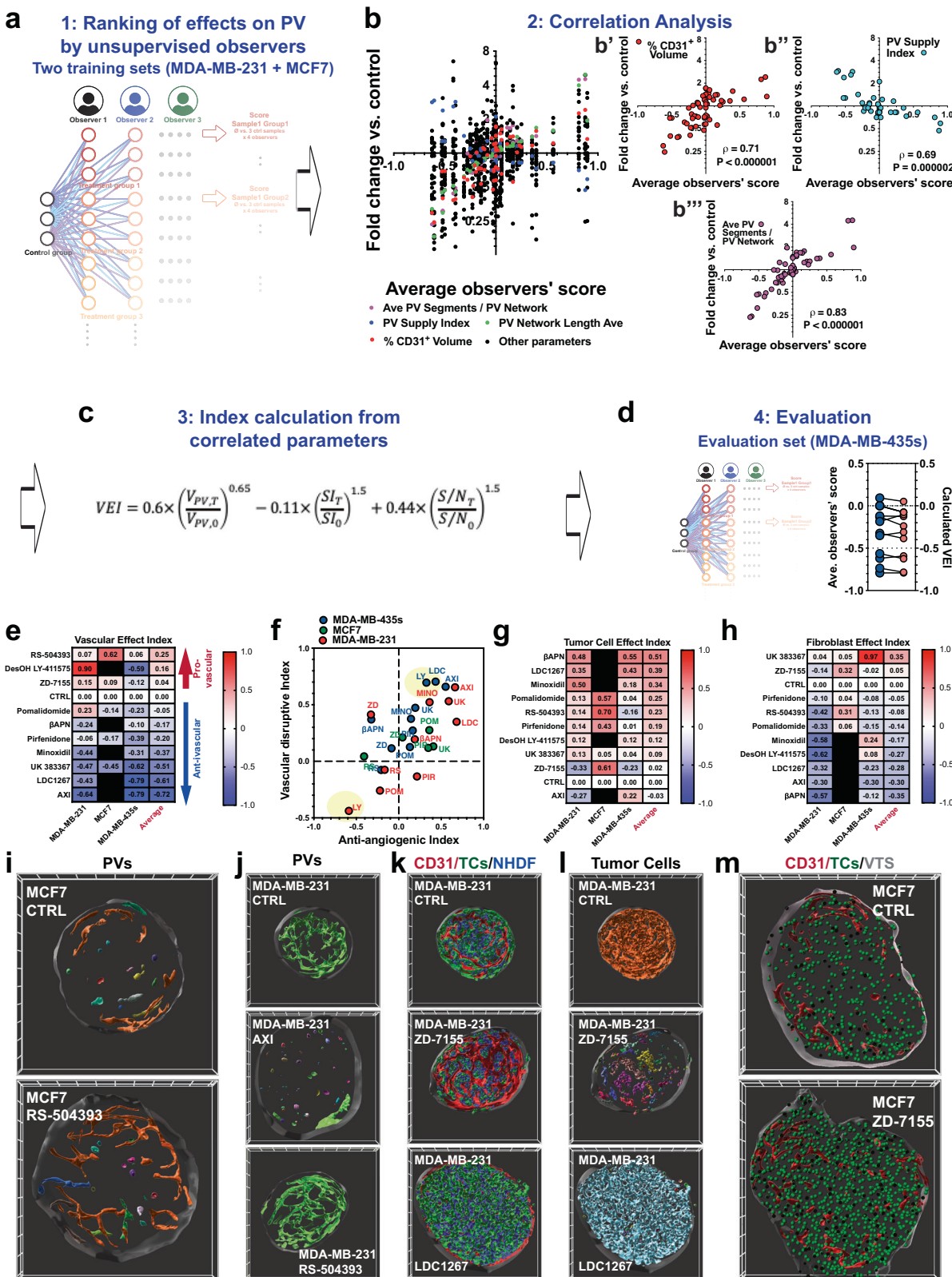

$$VEI = 0.6 \times \left(\frac{V_{PV,T}}{V_{PV,0}}\right)^{0.65} - 0.11 \times \left(\frac{SI_T}{SI_0}\right)^{1.5} + 0.44 \times \left(\frac{S/N_T}{S/N_0}\right)^{1.5}$$

detailed analysis of drug effects. This results in a high discriminative power, which is necessary for target selection, thus choosing candidates for further development out of a pool of verified, related targets. Secondly, the detailed output also allows for the uncovering of drug effects that manifest only in moderate and easy-to-miss changes of individual parameters but that shift several connected aspects in parallel. In combination, this can indicate significant alterations that would be missed by looking at individual parameters. Third, the comprehensive acquisition of parameters from the high-resolution, multichannel 3D datasets enables the discovery of unanticipated drug effects that might nevertheless prove interesting and desirable. Finally, the highly detailed analysis provides information that might look gratuitous at the time of analysis, but that can be saved and reassessed in the future with respect to a newly emerged therapeutic goal.

**Fig. 6 | Ranking of drug effects on VTS composition and architecture. a** Four untrained observers were tasked with rating the PV in all treated VTSs in the MCF7 and MDA-MB-231 sets against all adequate controls. The observer score (OS) represents the average of all ratings. **b** Correlation analysis of the OS for all treatments in the MCF7 and MDA-MB-231 sets versus numerical PV parameters. Significant correlation was found for changes in relative CD31+ volume, PV supply index, and in number of PV segments/PV network. Results from two-tailed correlation analysis with C.I. = 95%, and calculated Pearson correlation coefficient. **c** Equation with calculated factors for correlation of numerical PV parameters with OS. **d** Comparison of OS versus calculated vascular effect index (VEI) in the MDA-MB-435s set used for evaluation. **e** Calculated VEIs of all tested drugs in the MCF7, MDA-MB-231, and MDA-MB-435s sets. **f** Anti-angiogenic versus the vascular disruptive effect of the tested drugs in the MCF7, MDA-MB-231, and MDA-MB-435s sets. Only the average of three samples is shown for clarity. **g** Calculated tumor cell effect index of all tested drugs in the MCF7, MDA-MB-231, and MDA-MB-435s sets. **h** Calculated fibroblast effect index of all tested drugs in the MCF7, MDA-MB-231,

and MDA-MB-435s sets. **i** Architecture of PV in MCF7 VTSs in control and RS-504393-treatment groups. 3D-rendering of CD31+-PV surfaces pseudo-colored to represent the different structural objects. Frontal cut through VTS center in the xy-plane. **j** Architecture of PV in MDA-MB-231 VTSs in control and treatment groups (AXI, RS-504393). 3D-rendering of CD31+-PV surfaces pseudo-colored to represent the different structural objects. Frontal cut through VTS center in the xy-plane. **k** Composition of MDA-MB-231 VTSs, control, ZD-7155, and LDC 1267 treatment groups. 3D-rendering of CD31+-PV surfaces (red), tumor cells (green), and fibroblasts (blue). Frontal cut through VTS center in the xy-plane, 50 μm depth. **l** The same frontal cut through treated MDA-MB-231 VTSs as in (**k**), but 3D-rendering of tumor cell clusters are pseudo-colored to represent different connected objects. **m** Frontal cut through MCF7 VTSs in the control group and after treatment with ZD-7155. 3D-rendering of CD31+-PV surfaces (red), and individual tumor cells (green) represented as 10 μm orbs. Frontal cut through VTS center in the xy-plane, 50 μm depth. 3D grid spacing: 50 μm, error bars: ± SEM, *n* = 3 individual biological samples. Source data are provided as a Source Data file.

Less complex, targeted drug screening assays designed for fast throughput leave, in general, only limited possibilities to identify drug effects outside the planned line of inquiry.

We deliberately used established tumor cell lines to generate VTSs for the drug testing platform. Significant advances have been made in the last years in establishing 3D-organoid cultures from patients[56,57]. Although these patient-derived organoids hold immense promise for personalized medicine, e.g., testing which drugs promise the best results in the treatment of an individual patient, the challenges in these primary cultures are notably different from those in establishing a screening system for drug candidates. The natural variance between individual patient-derived tumor samples precludes standardization of assay conditions. Established tumor cell lines, in combination with pre-screened stromal cells, on the other hand, can be the basis of a highly standardized platform, as we demonstrated.

As anticipated, the use of different established mammary tumor cell lines gave rise to VTSs with evidently distinct structural compositions. This was highly reproducible and again underlined the previous finding that tumors create their individual microenvironment as much as the TME, in response, determines the characteristics of the tumor cells[58,59]. The architecture and composition of the TME in the VTSs in our experiments showed distinct characteristics also found in breast carcinomas and other tumors: In several VTSs, tumor cells formed isolated, compact islands encapsulated by lanes of fibroblasts. In these models, pseudovessels run within the fibroblast lanes, avoiding contact with tumor cells, representing a histological feature of clear separation of tumor cells and stroma found in many carcinomas. In other VTSs, the various cell types appeared more stochastically mixed, with pseudovessels getting in direct contact with tumor cells – a progressed, invasive phenotype in breast cancer[60]. However, histologically, all VTS models described here are associated with a poorly differentiated grade 3 phenotype, characterized by high mitotic count and a lack of any tubule formation[61]. It must be mentioned that in breast carcinomas, histological characteristics are not correlated with molecular subtypes that predict response to certain forms of therapy (e.g. response to hormonal or anti-HER2 therapy)[60,62]. In summary, our VTSs recreate a span of histological features also found in patient breast tumors and the range of molecular subtypes of breast carcinomas. However, none of our VTSs recreates the TME of a specific breast carcinoma subtype.

The building blocks of a TME provided in our setup are essential. Sachs et al. reported that in establishing a biobank of primary breast cancer organoids from biopsies, only 45% of the material derived from ductal and around 30% of the material from lobular tumors formed solid spheroids[24]. In our hands, the co-cultivation with fibroblasts enabled the formation of spheroids with a reliability of 100%, even from tumor cell lines that do not form spheroids by themselves. Aside from providing structural support directly and in the form of

ECM-secretion, fibroblasts mediate cell-cell interaction in the TME and provide important cues. Among others, fibroblasts have been shown to be essential in angiogenesis. Fibroblast-derived factors seem to be especially necessary for lumen formation[63], and beginning lumen formation was observed in our VTSs. Thus, the PV in the VTSs already shows important steps of maturation. Indeed, the VTS model we established, despite its artificial conception, mimics various complex features of the tumor/TME interaction: separation in a stromal and a tumor cell compartment and formation of a potential perivascular macrophage niche. Incorporation of other immune cells than macrophages that are routinely found in the TME could further improve the reliability of observed drug response and would enable the testing of other drug classes like immunotherapeutics. However, most immune cells are intolerant to prolonged in vitro cultivation, even in an engineered 3D-tissue setup. Additional basic research is needed to make these cells accessible for consistent drug screening.

Importantly, depending on which tumor cell line was incorporated, structural features were reproducibly altered, opening the possibility of modeling different distinct TMEs observed in tumors. The high reproducibility of this feature is witness to the ability of the cells to self-organize in our straightforward confrontation setup. Also, the structurally distinct VTSs from different tumor cell lines react differently to treatment with the various agents we tested. Generally, the effects of specific drugs went in the same direction in VTSs from different cell lines. However, the degree of response was significantly different. For instance, MDA-MB-231-based VTSs were much less sensitive to treatment with anti-angiogenic drugs than those created with MDA-MB-435s cells. A major future challenge will be to correlate this differential response to specific patient subsets. That is, to validate which histological or molecular markers in a patient's tumor decide which model VTS predicts with the most accuracy its response to a tested drug. This would not lead to personalized treatment in its strictest sense but would enable better stratification that could improve treatment outcomes substantially. Developing an approach that allows the inclusion of patient-derived tumor cells and possibly also stromal cells from the same tumor, based on our methods, seems possible. This would truly open the window for personalized testing of various treatment combinations. By basing the VTS on patient-derived material, a central goal is to a faithful recreation of the native tumor's TME. Therefore, criteria have to be established that allow for a quantitative assessment of how realistic the reproduction is. Moreover, patient tumor and stromal cells must be separated, pre-cultivated, and preconditioned to enable a successful reassembly. Endothelial and immune cells possibly have to still be provided from conventional cell culture, unless novel methods are developed to also cultivate them from native tumor material. In sum, the generation of microcopies of an actual tumor will involve a complex procedure that also has a high risk of creating significant artifacts that can alter drug response,

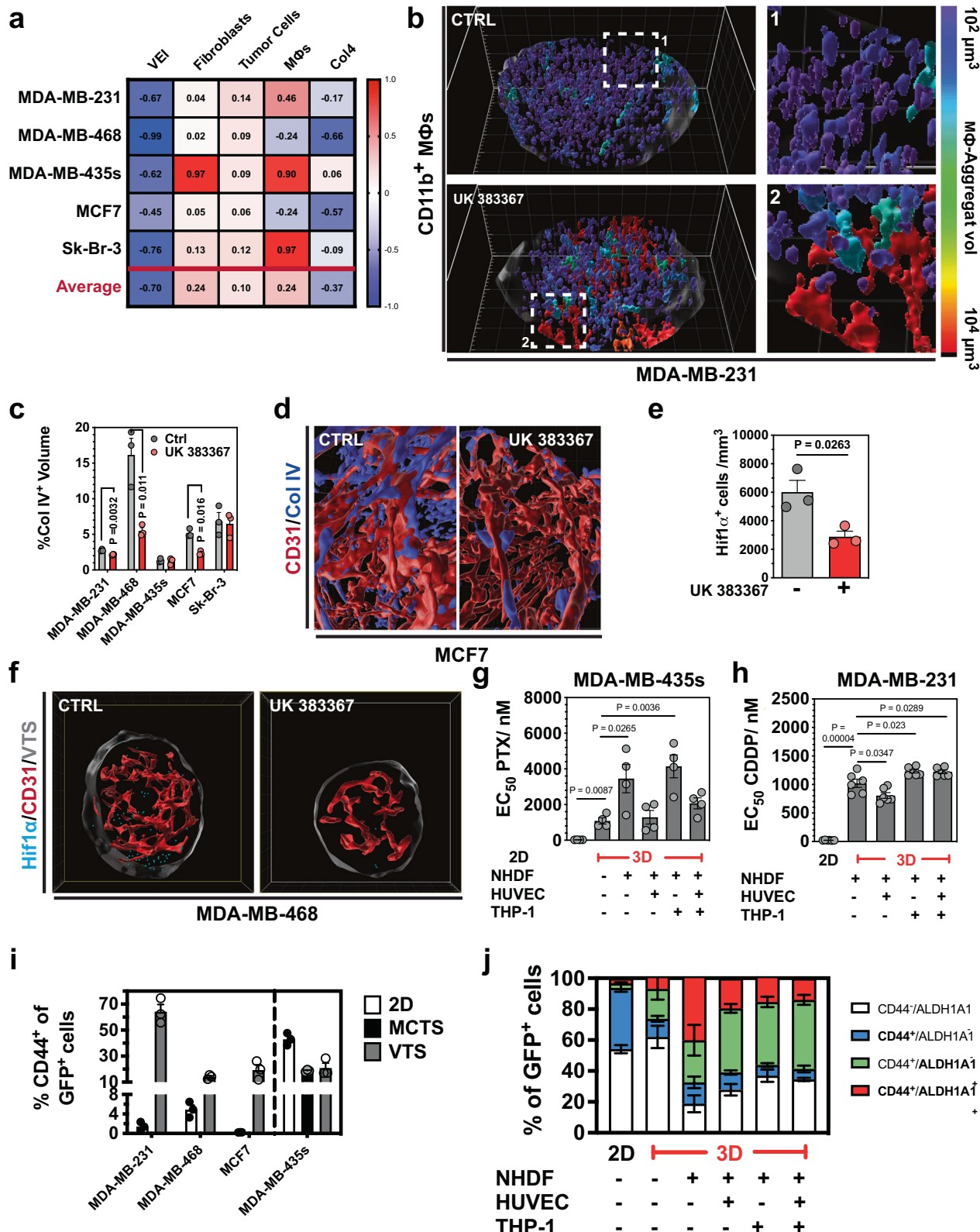

thereby jeopardizing the utility of such a system for personalized drug testing.

Designing assay systems and read-out methods that not only provide straightforward yes/no answers but allow for a ranking of drug candidates still poses a major challenge[64,65]. Multiparametric descriptive datasets like the ones obtained from the detailed structural analysis in this work enable comparison of drugs' effects with high discriminatory power. However, they do not per se yield benefits over simpler single-endpoint assays in enabling the ranking of drugs according to a desired effect. Yet, unlike single-endpoint assays, multiparametric descriptive datasets can contain the information for a meaningful ranking due to their richness of detail. By taking advantage of the intuitive assessment human observers are capable of, we developed a method to extract and weight the parameters from the

**Fig. 7 | Effects of the BMP-1 inhibitor UK 383367. a** Calculated effect indices on different compartments in UK 383367-treated VTSs. **b** Surface rendering of CD11b+-macrophages in control and UK 383367-treated MDA-MB-231 VTSs. Surfaces are pseudo-colored according to the volume of the connected macrophage aggregates. Quaternary cut through the equatorial and median plane. **c** Relative collagen IV+-volume in VTSs based on different tumor cell lines in control and UK 383367-treated VTSs. **d** CD31+- (red) and collagen IV+-volume (blue) in control and UK383367-treated MCF7 VTSs. Displayed is only the CD31-associated collagen IV+-volume. **e** Density of Hif1α+-cells in control and UK383367-treated MDA-MB-435s VTSs. **f** Distribution of Hif1α+ cells within VTSs generated with MDA-MB-435s cells. Hif1α+ cells are shown as pale blue orbs of 10 μm diameter, red: CD31+ PVs. Frontal cut through VTS center in the xy-plane, 100 μm depth. **g** EC50 values of PTX on

MDA-MB-435s cells in various microenvironmental contexts. $n = 4$. **h** EC50 values of CDDP on MDA-MB-231 cells in various microenvironmental contexts. $n = 6$. **i** Percent of CD44+ of all GFP+ tumor cells cultivated in either 2D or in a full VTS context with NHDFs, HUVECs, and THP-1. MDA-MB-435s were also cultivated as MCTS (spheroids without the addition of other cell types). **j** Results of FACS-analysis for ALDH1A and CD44 expression in MDA-MB-435s cells cultivated in either 2D or 3D spheroids, under the addition of different cell types. TCs were identified by their green fluorescence (GFP+), and ratios of ALDH1A+ and/or CD44+ cells within the GFP+ population were quantified. 3D grid spacing: 50 μm, error bars: ± SEM, analyzed with unpaired two-tailed t-test, $n = 3$, individual biological samples if not otherwise stated. Source data are provided as a Source Data file.

descriptive datasets most meaningful for vascular effects. This enabled a complete and particularized ranking of the drugs in our exploration field. The method can be used comparably for much larger datasets. Instead of a small set of observers, the task of ranking effects can be crowdsourced to a large field of observers. The power of such crowdsourcing approaches for medical image analysis has been thoroughly demonstrated[66]. Sufficiently large 3D-image sets annotated (=ranked) by crowdsourced observers could be used in a machine learning (ML) routine, aiming at identifying and ranking drug effects even more effectively. Large annotated datasets are fundamental for ML models, and our scoring system is a suitable basis for obtaining these annotations from crowdsourced observers[67,68]. Although the size of our image data set was still too small for an ML-based approach, a basic correlation analysis between observers' assessment and descriptive parameters already yielded a workable algorithm for drug effect ranking. The final aim is the establishment of a databank listing the effects of individual drug candidates in detail. Information can be extracted from this databank, in a ranked matter, at a later point according to emerging therapeutic goals. The high reproducibility of the self-organizing setup would allow for sequential addition to such a databank.

The drug evaluation platform is by design not intended for high-throughput screening of large substance libraries but more appropriate for validation and further characterization of hit compounds from a preceding, less complex screening procedure optimized for high throughput. The entire process of generation, treatment, harvest, staining, embedding, and clearing of VTSs can be run on current liquid-handling systems. Image processing/analysis and ranking of test compounds are automatable and scalable. Thus, throughput is limited by investment in computational power only. A limitation for throughput is the image acquisition by LSFM. To our knowledge, commercial solutions for automated sample acquisition in LSFM are currently not available. Robotic handling of hydrogel-embedded, cleared VTSs seems conceivable yet challenging, as problems like precise, unsupervised sample positioning in the light sheet need to be overcome. A simpler, less expensive solution might be the use of microfluidic systems to transport the processed VTSs, one after another, into the imaging window of the LSFM[69,70]. However, multichannel image acquisition by LSFM requires the VTS to be precisely kept in place and in the same orientation over several minutes. A constraint that is difficult to meet in microfluidic systems. Another possible approach to increase throughput is the use of chips that carry spheroids in individual wells that can be automatically placed in the lightpath by an appropriate stage[71].

Essential for the potential to implement our process into a larger screening industrial-scale setup is the observed robustness of the workflow in its entirety. Central to this robustness is the self-organizing capacity of the VTSs, which protects against outcome variability due to inevitable minor variations in experimental conditions. This robustness, in combination with the simplicity of the VTS-generation procedure, the potential for automatization, and the scalability of

computational processing tasks, opens the prospect for implementation as a test platform for target selection and drug evaluation.

## Methods
### Ethical statement
All experiments involving animals were reviewed and approved by the relevant regulation authority, the Regierung von Unterfranken, Würzburg (Germany). The experiments were performed in accordance with relevant guidelines and regulations.

### Chemicals and cell culture reagents
If not otherwise indicated, chemicals were purchased from Sigma-Aldrich (Munich, Germany) or Carl Roth (Karlsruhe, Germany), and cell culture reagents were acquired from Thermofisher (Darmstadt, Germany).

### Cell culture
MCF7 (HTB-22), MDA-MB-435s (HTB-129), MDA-MB-231 (HTB-26), Sk-Br3 (HTB-30), ZR75-1 (CRL-1500), and MDA-MB-468 (HTB-132) cells were obtained from ATCC. AT3 cells were obtained from Sigma-Aldrich (Catalog # SCC178). All tumor cells were maintained in DMEM (ThermoFisher, Catalog # 11995073) with 10% FBS and penicillin/streptomycin at 37 °C, 5% CO2. THP-1 (TIB-202) monocytes were obtained from ATCC and maintained in RPMI 1640 (ThermoFisher, Catalog # 11875093) with 10% FBS and penicillin/streptomycin at 37 °C, 5% CO2[72]. The identity of human tumor cell lines was validated by the cell line authentication service of the German Collection of Microorganisms and Cell Cultures GmbH (DSMZ, Braunschweig, Germany). NHDF were purchased from Lonza (Catalog # CC-2509) and cultivated in DMEM (Gibco) with 10% FBS, penicillin/streptomycin, 1% non-essential amino acid supplement (ThermoFisher), and ascorbic acid-2-phosphate (Sigma-Aldrich) at 37 °C, 5% CO2. HUVEC-2 was obtained from Sigma-Aldrich (Catalog # C-12208) and cultivated for up to 5 passages in EGM-2 media (Endothelial Growth Medium-2, Cat# CC-3162, Lonza, Switzerland). Cell lines were routinely tested at least every six months for mycoplasma contamination (Mycoplasma PCR detection Kit, Catalog # G238, ABM, Richmond, BC, Canada) and found negative for mycoplasma contamination throughout the experiments.

### Multicellular tumor spheroids (MCTS)
Tumor spheroids were generated by the liquid overlay technique, using the protocol from Walser et al. with modifications[73]. Wells of a 96-well culture plate were coated with 50 μL of 1.0% agarose in water. After the agarose had solidified, 2000 tumor cells were seeded in 200 μL medium (DMEM (ThermoFisher, Catalog # 11995073) with 10% FBS and penicillin/streptomycin) on top of the coating. Cells were incubated at 37 °C, 5% CO2 for six days with an exchange of 50% media volume on days 4, 7, and 10. Media was exchanged by slowly aspirating 100 μL of media at the side of the culture well with a pipette before 100 μL of fresh media was added. Alternatively, tumor cells were mixed with fibroblasts at various ratios: The cells were seeded at 2000 tumor cells/well with the respective fibroblast numbers in 200 μL medium.

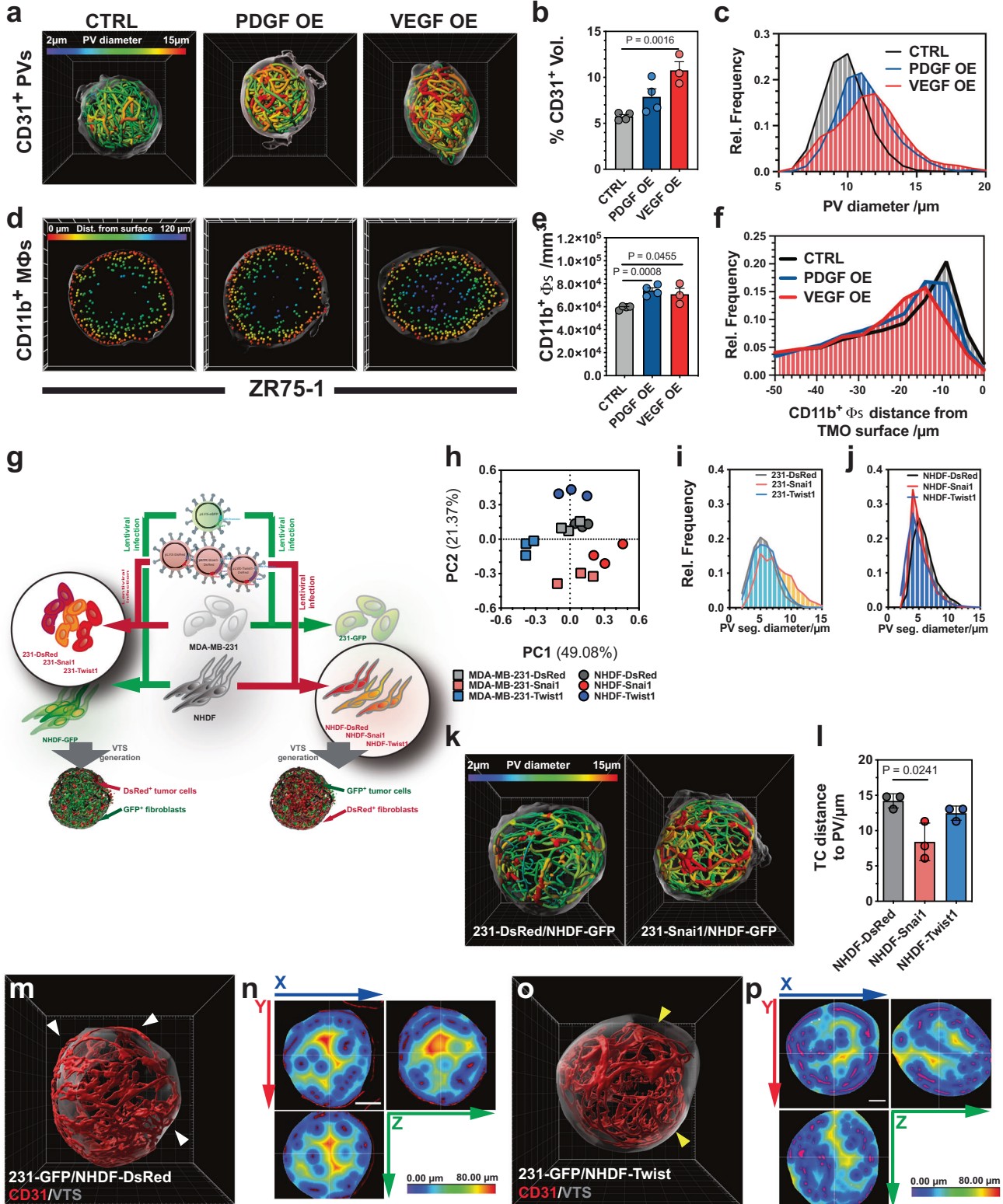

## Antibodies

Antibodies used for immunofluorescence staining: anti-CD31 (Dako, Mouse, Cat# M0823, RRID AB_2114471), anti-CD11b (Sigma-Aldrich, Rabbit, Cat# SAB5600105, RRID:AB_2910138), anti-Collagen IV (Bio-Rad, Rabbit, Cat# 2150-0140, RRID:AB_2082644), anti-Hif1α (Bethyl (Thermofisher), Rabbit, Cat# A300-286A, RRID:AB_2117114), Cleaved-Caspase 3 (Cell Signaling, Cat# 9661, RRID:AB_2341188). Secondary antibodies: goat anti-mouse-Cy5 (Jackson ImmunoResearch, Cat# 115-175-166, RRID:AB_2338714), goat anti-rabbit-Alexa Fluor 750 (Thermofisher, Cat# A-21039, RRID:AB_2535710). Dilutions are listed in Supplementary Table 5.

Antibodies used for FACS-analysis: anti-ALDH1A1-PE (SinoBiological, Cat# 11388-MM03-P, RRID:AB_2860344) and anti-CD44-Alexa Fluor 700 (ThermoFisher, Cat# 56-0441-80, RRID:AB_494012).

**Fig. 8 | Genetic manipulation of defined cellular components. a** Traced PVs in VTSs generated from CTRL, VEGF-A, or PDGF-B OE ZR75-1 tumor cells. PV segments are color-coded according to mean diameter. **b** Percentage of CD31$^+$-volume of total VTS volume in ZR75-1 VTSs. CTRL and PDGF-OE: $n = 4$; VEGF-OE: $n = 3$. **c** Distribution of PV-segment mean diameters in ZR75-1 VTSs. CTRL and PDGF-OE: $n = 4$; VEGF-OE: $n = 3$. **d** Position of CD11b$^+$-macrophages in CTRL, VEGF-A, or PDGF-B OE ZR75-1 VTSs. Frontal cut through VTS center in the xy-plane, 50 µm depth. **e** Density of CD11b$^+$-macrophages in ZR75-1 VTSs. CTRL and PDGF-OE: $n = 4$; VEGF-OE: $n = 3$. **f** Distribution of minimal distance of CD11b$^+$-macrophages to the VTS surface in ZR75-1 VTSs. CTRL and PDGF-OE: $n = 4$; VEGF-OE: $n = 3$. **g** Schematic view of the generation of Snai1/Twist1 OE MDA-MB-231 tumor cells and fibroblasts to produce genetically modified VTSs. **h** Principal component analysis of effects from Snai1 or Twist1 OE in MDA-MB-231 tumor cells or NHDFs. Analysis based on a 49-parameter descriptive data set. **i** Distribution of PV-segment diameters within VTSs generated from CTRL, Snai1, or Twist1 OE MDA-MB-231 cells. Average of 3 VTSs/group. **j** Distribution of PV-segment diameters within VTSs generated from CTRL, Snai1, or Twist1 OE NHDF. Average of 3 VTSs/group. **k** Traced PVs in VTSs generated from CTRL or Snai1 OE MDA-MB-231. Color-coded according to PV mean diameter. **l** Average distance of tumor cells to nearest pseudovessel in VTSs generated from CTRL, Snai1, or Twist1 OE NHDF. **m** PV networks (red) within a VTS (surface: transparent gray) generated from MDA-MB-231-GFP and NHDF-DsRed. Some PVs are located close to or on the surface of the VTS (white arrowheads). **n** Heatmap, displaying distances from the nearest PV (red) in a VTS generated from MDA-MB-231-GFP and NHDF-DsRed. The VTS's center is almost void of PVs. **o** PV networks (red) within a VTS (surface: transparent gray) generated from MDA-MB-231-GFP and NHDF-Twist1. The outermost layer of PVs is substantially retracted from the surface of the VTS (yellow arrowheads). **p** Heatmap, displaying distances from the nearest PV (red) in a VTS generated from MDA-MB-231-GFP and NHDF-Twist1. PVs also pervade the center of the VTS. 3D grid spacing: 50 µm, scale bars: 100 µm error bars: ± SEM, analyzed with unpaired two-tailed $t$-test, $n = 3$, individual biological samples if not otherwise stated. Source data are provided as a Source Data file. OE overexpression.

## Production of lentiviral particles and generation of ectopic protein overexpressing tumor cells and NHDF

ORFs encoding proteins for OE were cloned into pLVX-DsRed-IRES-puro (Clontech, Mountain View, CA). Lentiviral particles were generated in HEK 293T cells by co-transfection of the lentiviral vector with the pCMV-ΔR8.9 and pCMV-VSV-G plasmids (both obtained from Addgene, Cambridge, MA), using a standard CaCl$_2$-based transfection method[74]. Supernatant collected from the HEK 293T cells 48 h post-transfection, was passed through a sterile 0.45 µm syringe filter and used to transfect tumor cells after the addition of sterile polybrene-solution (Hexadimethrinbromid, Sigma-Aldrich, 8 µg/mL final conc. in supernatant). This procedure was repeated once 24 h later. Stable cells were selected by treatment with puromycin (5 µg/mL) for 48 h.

## Vascularized tumor spheroids (VTS)

VTSs were generated by the liquid overlay technique: wells of a 96-well culture plate were coated with 50 µL of 1.0% agarose in water. After the agarose had solidified, cells were seeded in 200 µL EGM-2 media (Endothelial Growth Medium-2, Cat# CC-3162, Lonza, Switzerland) on top of the coating. Plates were incubated at 37 °C, 5% CO$_2$ for nine or twelve days with an exchange of 100 µL media on days 4, 7, and 10. Media was exchanged by slowly aspirating 100 µL of media at the side of the culture well with a pipette before 100 µL of fresh media was added. Standard seeding densities (if not otherwise indicated) were 100 THP-1, 1,000 HUVEC, 2000 tumor cells, and 20,000 fibroblasts per well. When working in the 96-well format, only the inner 60 wells (wells B2-G11) were used for VTS generation, and the outer rim of wells was filled with 250 µL of sterile PBS.

## Drug treatment of MCTS and VTS

VTSs/MCTS were exposed to drugs by exchanging 100 µL of media containing the agent at 2.5-fold the desired final concentration to adjust for the total volume of liquids in the well (200 µL media + 50 µL agarose). For treatment periods longer than three days, media was again exchanged every three days with fresh media containing the desired final concentration of the agent. Agents were not removed until termination of the cultivation and harvest of the VTSs/MCTS.

## Harvest and fixation of VTSs.

VTSs were directly extracted from the culture well in a volume of 80 µL of media by centering a large bore 200 µL pipette tip within the well right above the agarose layer and rapidly aspirating the media. The media with the VTS was released in a standard 1.5 mL microreaction vessel (MRV). The successful extraction of the VTS was directly checked by microscopic observation of the well at low magnification. 10–15 VTSs were collected in one MRV and subsequently processed together.

The VTSs were allowed to settle by gravity in the tip of the MRV – a process that takes with most VTSs less than a minute. The media was

aspirated, and the VTSs were washed by applying 1 mL PBS, letting the VTSs settle, and removing the supernatant. The VTSs were fixed for 1 h at r.t. in 4% PFA in PBS, then to an ascending row of diluted MeOH: 50% MeOH for 20 min, 80% of MeOH for 20 min, and 100% MeOH for 10 min. The MeOH was replaced with fresh MeOH, in which the VTSs could be stored for prolonged times at −20 °C.

## Immunofluorescence staining of VTSs.

The following buffers were prepared and used in the staining procedure: (i) Penetration Buffer: 25% (v/v) 1.5 M glycine in water, 0.25% (v/v) Triton X-100 in PBS. (ii) Washing Buffer: 0.2% (v/v) Tween-20 in PBS. (iii) Blocking Buffer: 10% (v/v) DMSO, 6% (w/v) BSA, 0.2% (v/v) Tween-20 in PBS. (iv) Antibody Buffer: 5% (v/v) DMSO, 3% (w/v) BSA, 0.2% (v/v) Tween-20 in PBS.

If not otherwise indicated, all washes and incubation steps were performed at r.t. The incubation intervals given should be considered as minimums, and can be increased within reasonable extents but should not be shortened. All steps were performed in 1.5 mL MRVs. In general, 1 mL of the respective buffer was applied, and the stirred-up VTSs were allowed to settle by gravity before the buffers were removed stepwise with a 200 µL pipette. In most steps, the incubation times are more than enough for the VTSs to settle.

VTSs stored in MeOH were rehydrated in changes of 2 × 20% DMSO in MeOH for 20 min each, 1 × 80% MeOH for 10 min, 1 × 50% MeOH for 10 min, 2× PBS for 10 min, and 2 × 0.2% (v/v) Triton X-100 in PBS for 10 min ea. To improve antibody (AB) penetration, the VTSs were incubated for 60 min in penetration buffer, then for 60 min in blocking buffer before being washed twice in washing buffer for 60 min each. The primary ABs were applied in a small volume of 50 µL of antibody buffer at an appropriate dilution and incubated ON at 4 °C. The primary ABs were removed by 6 washes with washing buffer for 30 min ea. The fluorophore-labeled secondary ABs were again applied in a small volume of app. 50–100 µL of antibody buffer at an appropriate dilution and incubated ON at 4 °C. The ABs were again removed by 6 washes with washing buffer for 30 min each. Finally, the VTSs were washed once in 1 mL water. This is necessary to remove NaCl, which would otherwise impede the setting of the agarose.

## Hydrogel embedding and optical clearing.

A heating block was warmed to 55 °C, and the MRVs containing the washed VTSs were inserted and left to equilibrate. 1.2% agarose in water was (re)-melted by boiling. Agarose was transferred into 1.5 mL MRVs and also placed into the heating block to adjust to the temperature of 55 °C. After ca. 10 min, the temperature-equilibrated agarose was applied: 100 µL were carefully pipetted on top of the VTSs by letting the agarose slowly flow down the inner MRV wall. The aim is not to stir up the VTSs such that they remain concentrated in one spot at the outer side of the tip of the forming plug. If the VTSs get detached, they can be re-collected by a brief spin in a microcentrifuge (set at r.t.) immediately after application

of the agarose. The agarose was allowed to set at r.t. Afterward, the hydrogel plug is overlaid with 1 mL of 50% EtOH in 10 mM Tris HCl, pH 9.0 for 1 h at r.t. To proceed with dehydration, the overlaid liquid phase is subsequently changed to 70% EtOH, 90% EtOH, and two changes of 96% EtOH (i.e., technical grade EtOH) each for 1 h. At this point, the dehydrated agarose gel is much sturdier than in its water-based form and has shrunken a bit. Thus, it can easily be removed from the MRV by flipping the MRV's tip while holding it upside-down over a 10 mL snap-lid glass vial. This can be facilitated by releasing the gel with a syringe needle or a *small* spatula. After the gel plug is transferred to the glass vial (the embedded VTSs are often nicely visible at this stage), the plug is overlaid with 5–10 mL EtOH$_{abs}$ for 4 h at r.t. The EtOH$_{abs}$ is exchanged one more time and left ON. The change to EtOH$_{abs}$ (not technical grade EtOH!) is necessary to remove all traces of water. Otherwise, the plug shrinks and withers after application of the ethyl cinnamate. After the plugs are thoroughly dehydrated, they are overlaid with 5 mL of ethyl cinnamate (ECi) for 4 h. The plugs first float but sink within a short time. At this point, the plugs should have turned "invisible" within the solvent as a consequence of the matched r.i. ECi was changed one more time and stored at least overnight before imaging. The plugs containing the fluorescence-labeled VTSs can be stored in ECi for at least 3 months at 4 °C in the dark.

## Image acquisition

Cleared hydrogel-embedded VTSs were imaged at a custom-built light sheet fluorescence microscope with a 20× objective (HCX APO L 20×/0.95 IMM; Leica, Mannheim, Germany) and recorded as image stacks with a pixel size of 0.5 μm/px and a z-step distance of 1 μm. The hydrogel plugs were attached to the head-mounted xyz stage (8MT167-25 XYZ 3-axis translation system, Standa, Vilnius, Lituania, controlled via a TANGO 3 PCI-E card, Märzhäuser, Wetzlar, Germany) using a pincer clip and submerged in the ECi-filled imaging chamber (Supplementary Fig. 11a). The custom-built LSFM setup can be described as follows (see also Supplementary Fig. 11b): a customized fiber-coupled laser combiner (BFI OPTiLAS GmbH, Groebenzell, Germany) provided the required excitation lines of 491, 532, 642, and 730 nm. For laser beam collimation, two objectives (RMS10X-PF Thorlabs, Bergkirchen, Germany) for VIS and 730 nm were used. A DCLP 660 dichroic beam splitter (AHF Analysentechnik, Tübingen, Germany) combined the two beam paths. A following telescope (BEX 1×-4×017052-202-26, Jenoptik, Jena, Germany) served to adjust the beam diameter. Alternating dual-side illumination was realized by a two-axis galvanometer scanner (6210H; Cambridge Technologies, Bedford, MA, USA) in combination with a theta lens (VISIR f. TCS-MR II; Leica, Mannheim, Germany) which finally created a virtual light sheet that was additionally pivot scanned by a single-axis resonant scanner system (EOP-SC, 20-20×20-30-120; Laser2000, Wessling, Germany) to minimize shadowing artifacts. The light sheet was projected onto the sample via a 200 mm tube lens (TTL200, Thorlabs, Bergkirchen, Germany) and a lens objective (Nikon CFI60 TU Plan Epi 5×/0.15, Edmund Optics, York, United Kingdom). The objective on the detection side (HCX APO L 20×/0.95 IMM; Leica, Mannheim, Germany) placed on a piezo positioning system (P-611.1 and E-665, PI, Karlsruhe, Germany) for focus correction collected the fluorescence perpendicularly to the light sheet, and, in combination with an infinity-corrected 1.3× tube lens (model 098.9001.000; Leica, Mannheim, Germany), projected the image into a scientific complementary metal oxide semiconductor (sCMOS) camera (Neo 5.5; Andor, Belfast, United Kingdom) (2560 by 2160 pixels, 16.6-mm-by-14.0-mm sensor size, 6.5-μm pixel size). The fluorescence was spectrally filtered by typical emission filters (Semrock, IDEX Health & Science, Rochester, NY) according to the use of the following fluorophores: BrightLine HC 525/50 (Alexa Fluor 488 or autofluorescence), BrightLine HC 580/60 (Alexa Fluor 532), HQ697/58 (Alexa Fluor 647), BrightLine HC 785/62 (Alexa Fluor 750). Filters were part of a motorized filter wheel (MAC 6000 Filter Wheel Emission TV

60C 1.0× with MAC 6000 controller; Zeiss, Göttingen, Germany) placed in the collimated lightpath between the detection objective and the tube lens. (For alternative setups of LSFMs see[75,76]).

Hardware components for image acquisition (laser, camera, stage, filter wheel) were controlled by Andor IQ 2.9 software (Oxford Instruments Abington UK, formerly Andor Technologies, Belfast, UK).

## Image processing and analysis

All processing and analysis steps were performed on a workstation equipped with two Intel Xeon Gold 6244 8-core-CPUs (3.6 GHz), 1024 GB RAM, and an NVIDIA Quadro RTX 6000 (24 GB) graphic card using the indicated software.

**Pre-processing.** The acquired multichannel image stacks were loaded into the Fiji distribution of ImageJ[77]. To reduce noise, channel stacks were processed with a median 3D filter (setting 2.0 in x, y, and z directions). Afterward, an additional channel was generated by additively combining all acquired channels. This additional channel was subsequently used to model the surface of the VTS. The processed stack was saved as a single TIF file and converted using the Imaris file converter. Converted files were read into Imaris (Version 9.6.0 - 10.0, Oxford Instruments, Abington, UK).

**Segmentation of cellular and molecular structures.** Surfaces were calculated for each channel. A surface detail of 2 μm was chosen. Threshold and filter settings in the routine were set individually for each acquired set of stacks according to image intensity and quality. These parameters were left unchanged for the processing of the whole set of stacks. Between sets of stacks that were imaged at different time points, only thresholds for signal/background distinction were adjusted, to accommodate for differences in staining intensity and illumination during imaging. To model the surface of the VTS, the additional, combined channel was selected and processed with a surface detail of 10 μm. Standardized settings were used for surface segmentation (Supplementary Table 6). These standardized settings were saved within Imaris for batch processing and subsequently used to analyze all stacks within a set.

**Distance transformation.** The *Matlab-based* (MathWorks, Natick, MA) distance transformation routine in Imaris was used to first visualize the distance of individual voxels to the surface of the nearest PV structure. This generated a new 8-bit channel in the 3D stack in which the distance to the PV surface was decoded by a gray value in the range of 0–255. Masking the result to the volume inside the VTS was then realized by setting all voxels in the new channel outside the VTS surface to a black value of 255. The now masked channel was exported to Fiji. Using the histogram of the 3D stack allowed the export of the visual data into numerical values for further analysis.

**Tracing of pseudovascular (PV) networks.** For tracking, the filament add-in in Imaris 10.0 was utilized. Based on the CD31-channel the PV structures were traced and combined into connected networks. Standardized settings were used for generating traces (Supplementary Table 7).

**PV heterogeneity.** The heterogeneity of the PV network within the VTSs was indexed by the fraction of the volume percentage of the CD31$^+$ PV within the VTS and the median distance of each voxel outside the PV volume but inside the VTS.

**Calculating vascular effect index.** Observers were provided with 3D representations of the PV in VTSs and tasked with rating PV networks after treatment in a side-by-side comparison to an untreated control from the same experiment, according to a 5-grade scheme: +2 (strongly improved), +1 (improved), 0 (unchanged), −1 (impaired), −2

(strongly impaired). Each treated VTS was compared to all three or four control VTSs from the experiment. Four observers ranked all treated VTSs, and an average of all ratings (4 observers × 3 (4) ratings versus controls × 3 treated VTSs) was assigned as an observer score (OS).

The OS was analyzed for correlation with parameters describing the PV network using Prism 10 (GraphPad, LaJolla, CA). A statistically significant correlation was calculated for parameters describing the absolute volume ($V_{PV}$), the distribution (supply index (SI)) and the number of segments per connected network (S/N) of the pseudo-vasculature. Thus, observers tended to relate the quality of the network with its overall size, its homogeneity within the VTS, and its complexity.

The OS was used to calculate a vascular effect index (VEI) to allow for rapid estimation from numerical parameters of how observers would rank a PV network in comparison to a non-treatment control. As $V_{PV}$, SI, and S/N correlated with OS, it was expected that an equation of the following form would allow calculating an approximate VEI:

$$VEI = x_1 \times \left(\frac{V_{PV,T}}{V_{PV,0}}\right)^{y_1} + x_2 \times \left(\frac{SI_T}{SI_0}\right)^{y_2} + x_3 \times \left(\frac{S/N_T}{S/N_0}\right)^{y_3} \quad (1)$$

With:

$x_1, x_2, x_3$: weight factors
$y_1, y_2, y_3$: non-linear correction factors

The respective parameters ($V_{PV}$, SI, and S/N) and OS from 17 treatment groups involving MCF7- and MDA-MB-231-based VTSs were used to generate a system of non-linear equations. Using Matlab (version R2021a, MathWorks, Natick, MA), approximations for $x_1$, $x_2$, $x_3$, $y_1$, $y_2$, and $y_3$ were obtained in a stepwise process estimating first the weight factors ($x_1$–$x_3$), then the exponents correcting for a non-linear correlation ($y_1$–$y_3$). Weight factors and exponents were

$$VEI = 0.6 \times \left(\frac{V_{PV,T}}{V_{PV,0}}\right)^{0.65} - 0.11 \times \left(\frac{SI_T}{SI_0}\right)^{1.5} + 0.44 \times \left(\frac{S/N_T}{S/N_0}\right)^{1.5} \quad (2)$$

For evaluation VEIs were calculated for treatment groups with MDA-MB-435s-based VTSs and compared to the respective OSs.

## Cell toxicity assay

VTSs/MCTS were generated with MDA-MB-231 (231-luc) or −435s cells (435s-luc), engineered by lentiviral transfection to express firefly luciferase. The vector used for generating lentiviral particles to transfect the parental lines was pLVX-Puro-Luc. After infection, cells were selected with 5 ng/mL puromycin for 48 h.

Generation of VTSs/MCTS was performed in sterile white 96-well MWDs coated with 50 µL sterile agarose (1% in water). 231-luc or 435s-luc were mixed at the previously established ratio (THP-1: TCs: HUVEC: NHDF = 100: 1000: 2000: 20,000 in 200 µL media) with some or all the stromal cells and the mixture overlaid on the agarose. After 6 days of cultivation, 100 µL of media was exchanged with 100 µL fresh media containing the appropriate cytotoxic agent (Cisplatin for 231-luc, paclitaxel for 435s-luc) at the 2.5x final concentration. The agents were applied at an 8-step gradient (0 nM, 61.7 nM, 185.2 nM, 555.6 nM, 1.667 µM, 5 µM, 15 µM, 45 µM final concentration) each in six replicates. Cells were exposed to the drug for 72 h. To measure the luminescence of remaining tumor cells, 100 µL media was removed, and 50 µL/well (25 µL Buffer 1 followed by 25 µL Buffer 2) LUC-Screen Extended Glow Assay solutions (ThermoFisher, Cat. # T1035) were added. Luminescence was recorded after 15 min of incubation at r.t. on a Wallac multimodal plate reader with luminometer function (PerkinElmer).

## Flow cytometry

VTSs were harvested by aspiration with a pipette. 10–20 pooled VTSs were washed with PBS and digested for 10 min in trypsin/EDTA in PBS at 37 °C then for 20 min in 10 mg/mL collagenase I (Worthington

Biochemical Co., Lakewood, NJ) in DMEM (w/o FBS or P/S) at 37 °C. The single-cell solution was washed twice with PBS, and resuspended in 1% FBS in PBS. Cells were stained for CD44 surface marker expression (anti-CD44-Alexa Fluor 700 antibody, Invitrogen Cat# 56-0441-80 RRID:AB_494012, dilution 1:200), while intracellular staining using BD Cytofix/Cytoperm Fixation/Permeabilization Kit (BD Bioscience, Heidelberg, Germany, Cat# 554714,) was applied for ALDH1A1 (anti-ALDH1A1-PE antibody, Sinobiological Cat# 11388-MM03-P RRID:AB_2860344, dilution 1:20). Dead cells were removed from the analysis using gating for negative cells after staining with a Live/Dead staining kit (Thermofisher, Cat# L34975). Probes were analyzed using a FACS Celesta flow cytometer and the FlowJo 10.0 software (BD Bioscience, Heidelberg, Germany,).

## Principal component analysis (PCA)

PCA was performed using OriginPro (OriginLab, Northhampton, MA). Datasets of >45 parameters descriptive of the VTSs architecture, composition, and distal relationship of cellular compartments were standardized along parameters and subjected to PCA. The number of parameters available for analysis varied in the various experiments, depending on which cellular compartments immunofluorescence staining was performed.

## Murine tumor model

All experiments involving animals were reviewed and approved by the relevant regulation authority (protocol #87/10), the Regierung von Unterfranken, Wrzburg (Germany). The experiments were performed in accordance with relevant guidelines and regulations.

The protocol, approved by the Regierung von Unterfranken, allows for a maximal tumor size of 1 cm³. This size was not reached or exceeded in the described studies.

**Engraftment.** AT3 ($1 \times 10^6$ cells in PBS) breast adenocarcinomas were generated by injection of cells into the inguinal mammary fat pad of 8-week-old female wild-type C57Bl/6J mice (CharlesRiver, Sulzfeld, Germany). Success of engraftment was assessed 7–9 days post-injection, and tumor-bearing mice were randomly assigned to the different treatment groups just prior to the start of treatment.

**Treatment.** Starting on day 15 post-injection, mice were treated for six days q.o.d. (that is, on days 15, 17, and 19) with the inhibitors at the following doses:

Axitinib 25 mg/kg BW q.o.d.; The appropriate amount of AXI was dissolved in 200 µL/dose 0.2% methylcellulose (Sigma-Aldrich, Munich, Germany) in sterile water by sonification.

UK 383367 was applied at 25 mg/kg BW q.o.d. The appropriate amount of UK 383367 was first dissolved in 30 µL/dose EtOH$_{abs}$ (10× concentration). This EtOH solution was diluted with 270 µL/dose of 10% (2-Hydroxypropyl)-β-cyclodextrin (Sigma-Aldrich, Munich, Germany) prior to injection.

2-Aminopropriotril was applied at 100 mg/kg BW q.o.d. The appropriate amount of βAPN was dissolved in 200 µL/dose of sterile PBS.

Control animals received the same amount of carrier solution (i.e. 200 µL 0.2%methylcellulose or 300 µL 10%EtOH in 10% (2-Hydroxypropyl)-β-cyclodextrin or 200 µL PBS). Drug solutions and control substances were applied by intraperitoneal injection.

On day 21 post-injection, animals were retro-orbitally injected with 10 µg Alexa-647 labeled CD105 antibody (ThermoFisher, Cat# MA1-19543, RRID:AB_1071127) in 50 µL PBS and sacrificed 20 min later by $CO_2$ asphyxiation. Euthanized mice were perfused through the heart with 10 ml PBS, followed by 10 mL of 4% (w/v) PFA in PBS. Organs and tumors were removed and submersion-fixed for 24 h at 4 °C in 4% (w/v) PFA in PBS. Excess PFA was removed with two changes of PBS for 24 h at 4 °C.

**Clearing.** For clearing the organs were first cut in ca 2–4 blocks and dehydrated in EtOH (50% EtOH in 10 mM Tris HCl pH 9.0 (4 h, 4 °C), 70% EtOH in 10 mM Tris HCl pH 9.0 (4 h, 4 °C), 90% EtOH in 10 mM Tris HCl pH 9.0 (4 h, 4 °C), 96% EtOH (4 h, 4 °C), 96% EtOH (4 h, 4 °C), 100% EtOH (100 h, 4 °C), 100% EtOH (4 h, 4 °C),) before being placed in excess of ethyl cinnamate (ECi) for at least 24 h at r.t. after which ECi was exchanged. Treatment in ECi continued until tumor blocks appeared transparent. The samples were kept afterward in ECi until imaging.

**Imaging.** Cleared tumors were placed in the lightpath within the ECi-filled objective chamber of the custom-made light sheet microscope described above using a pincer-holder. Stacks were acquired in increments of 1 µm by imaging each plane in two color channels (BrightLine HC 525/50 (autofluorescence), and HQ697/58 (Alexa Fluor 647)) sequentially. Typically, stacks of $1024 \times 1024 \times 400$–700 voxels with a voxel size of $0.5 \times 0.5 \times 1\,\mu m^3$ were acquired. Hardware components for image acquisition (laser, camera, filter wheel, stage, focus correction) were controlled by IQ 2.9 software (Andor, Belfast, United Kingdom). Images were saved as tagged image files (TIF), processed, and analyzed as described above for the processing of VTS image stacks.

## Statistical analysis

All samples were individual biological replicates, and no samples were repeatedly measured. Statistical analysis was done using the Prism9 Software (GraphPad, LaJolla, CA). Differences between the two groups were analyzed using an unpaired, two-tailed Student's T-test. In parallel, the samples were tested for significant variation of variance, and if necessary, a Welch correction was included in the statistical analysis.

## Reporting summary

Further information on research design is available in the Nature Portfolio Reporting Summary linked to this article.

# Data availability

The authors declare that the data supporting the findings of this study are available within the article and its Supplementary Information. Source data are provided with this paper.

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

## Acknowledgements

We gratefully acknowledge funding provided by the Interdisciplinary Center for Clinical Research at the Department of Medicine, JMU (IZKF, Grant Nos. Z-12 to J.P. and M.F. and B-369 to E.H.), the Deutsche For-schungsgemeinschaft (DFG, Grant Nos. HE3565/2-1, HE3565/3-1 to E.H., Project No. 324392634 TRR221 to A.Z., and Project No. 326998133 TRR 225 subproject Z02 to K.G.H. and K.H.), and the Wilhelm-Sander Foundation (Grant Nos. 2015.001.01 and 2018.080.01 to E.H.).

## Author contributions

D.A., M.Ba., M.Bu., D.SM., V.O., A.Ö., C.M. and E.H. performed the experiments. D.A. M.Ba., J.P., M.F., V.O., D.S-.M., V.O., C.M. and E.H. acquired the visual data. D.A. M.Ba., M.Bu., D.S.-M., C.M. and E.H. analyzed the data. D.A., M.H., S.E., K.H. and E.H. discussed the data. M.H., K.G.H. and E.H. developed the methodology. The experiments were planned by D.A., R.N., A.Z., M.H., K.G.H. and E.H. The manuscript was written by E.H. Before submission and during revision, K.H., M.Bu, K.G.H. and E.H. reviewed and edited the manuscript.

## Funding

## Competing interests

The authors declare no competing interests.
