## [Peer Review File · Nature Communications]

Reviewers' Comments:

Reviewer #1:

Remarks to the Author:

This manuscript developed a platform to generate tumor spheroids with different cell types and then implemented light sheet fluorescence microscopy (LSFM) to image and semi-quantitative analyze the spheroids. This combined platform/strategy provided many advantages over homogenous spheroids and other evaluation approaches. The authors further implemented the platform to screen and evaluate various antifibrotic and antiangiogenic drugs. The methods and approaches are innovative. But there are some concerns related to the verification of this platform, the similarity of generated spheroid model to the native tumor, and overall applicability for screen the drug for clinical usage.

1. The authors need to prove the accuracy of LSFM based analysis method. This is very important. The ratio of tumor:fibroblast was initiated from 1:20. The analysis of LSFM on day 0 should be presented.
2. How comparable is the generated TMO to native breast tumor tissues? The use of normal fibroblasts and THP-1 raises concerns
3. What medium was used for the culture of TMO? How the medium affect the TMO component changes.
4. One of the major concerns is that the whole analysis was based on LSFM. There was no other method was used to verify the results. For example, in Fig. 6, some other cellular/molecular assays should be used to verify the concept of vascular effect index.
5. There are many limitations and restrictions of the approach and platform. The authors should thoroughly discuss these points.

Minor:

1. The term "humanized" is not suitable to be used in this study.
2. The term "microorganoid" is also not accurate in this study.

Reviewer #2:

Remarks to the Author:

I co-reviewed this manuscript with one of the reviewers who provided the listed reports. This is part of the Nature Communications initiative to facilitate training in peer review and to provide appropriate recognition for Early Career Researchers who co-review manuscripts

Reviewer #3:

Remarks to the Author:

The authors present a platform for screening tumor organoids and demonstrate its utility with several cases studies.

The topic is of high importance, and the study seems to be thorough. However, after reading the manuscript several times, it remains unclear to me where exactly is the novelty of the results and what is the take-home message for readers. Perhaps it is due to the way results are structured and presented, but it remains vague. Is the novelty in the way of growing the specific micro-organoids? Is it in data analysis? Or in the findings of the case studies?

The manuscript's central topic is the new platform, but it is not clear how an interested reader can reproduce or implement the platform as a tool. The concrete actionable details to reproduce the

platform seem to be missing. Since I personally specialize in microscopy, below are my specific questions about this part.

Major

Methods, Image Acquisition, lacks many details that are expected:

- for adequate description of a custom microscope, a CAD model and sufficiently detailed technical drawings (2D and 3D, including the light paths, lenses, and general optomechanical arrangement) should be presented for readers to grasp the design principles used, if not to fully reproduce the design.

- l. 960-961, "custom-built light sheet microscope, with a 20x objective," - which objective? Manufacturer, model, NA, immersion type? I guess this is Leica HCX APO L 20x/0.95 IMM, introduced in 1.979, but it should be stated more clearly. I would either remove "with a 20x objective" altogether from the first sentence, or specify it fully.

- "The hydrogel plugs were attached to the head-mounted xyz stage using a pincer clip and submerged in the EtCi-filled imaging chamber." A photo or a drawing would be very helpful to visualize this.

- l.964 "A custom-built LSFM setup tailor-made for organ imaging"
Is it another setup, or the same one?

- what software was used to control the custom microscope(s)? Is this software already published? If not, do authors plan to publish it? If not published along with microscope design, it hardly counts as an adequate description of a platform.

Minor:

- MCTS abbreviation is introduced in line 156, but not explained until l. 837
- I would avoid double line spacing, it makes the text stretched and difficult to read.
- ethyl cinnamate is usually abbreviated as ECi, not EtCi
- antibodies dilution in Methods must be stated.

Clarity/presentation

"To guarantee undistorted 3D imaging, we decided to use light sheet fluorescence microscopy".
What exactly is undistorted imaging? Why does LSFM provide such "undistorted" imaging?

The text is often difficult to follow. For example, in the Discussion l.721: "Thus, throughput is limited by investment in computational power only. An exception is LSFM, where no commercial solution for automated sample throughput is available." I guess authors are trying to say that computational analysis is easily scalable (is it?), but high-throughput imaging is challenging, and light-sheet microscopy helps addressing this bottleneck. But the sentence is written ambiguously, and this is just one example.

To better structure the text, especially Introduction and Discussion, I recommend Mensh B, Kording K (2017) Ten simple rules for structuring papers. PLoS Comput Biol 13(9): e1005619.
<https://doi.org/10.1371/journal.pcbi.1005619>

Reviewer #4:

Remarks to the Author:

In this manuscript, the author presents results from an effort to promote the assembly of multiple cell types, including tumor cells, macrophages, endothelial, and fibroblasts, to form tumor microorganoids (TMO). They report a methodical microscopic analysis of the temporal steps in the formation of TMO and interesting computation renditions of the process. They propose that the use of established cell lines for TMO makes the system more robust and reduces the risks associated with using primary tumor materials. The authors also demonstrate the utility of the model to assess the effectiveness of therapies aimed at targeting cells in the microenvironment. Although

the study uses innovative imaging approaches, it lacks conceptual rigor and is too early to make impactful conclusions.

- 1) The authors argue that the failure of drugs in clinics is due to a lack of good models and claim that the TMO platform was developed to address the challenge, yet the studies presented are using systems that are unproven to model patient tumors, let alone addressing the challenge they propose to address.
- 2) Whether the platform has any potential for modeling clinical response is unclear. Most of the drugs used in the study have failed in breast cancer clinical trials, raising concerns about the utility of the platform.

In addition to the above-indicated major concern, there are multiple concerns:

- 3) The computational pseudo-vascular rendition images, while providing nice images, seem to have unclear biological significance as the authors do not demonstrate that the endothelia form a contiguous tubular structure that is capable of conducting fluids.
- 4) The authors have not supported the reasoning to believe the observed organization models anything relevant to breast cancers in patients. Given there is a great deal of patient-to-patient in the molecular and morphological organization of the tumors, it is not clear what the authors are modeling.
- 5) It is unclear if the co-culture impacts the molecular state of any cell types when they are part of the TMO? If being part of the TMO does not significantly alter the cell phenotype compared to the cells grown alone, then one may wonder if one needs to generate TMO to assess the efficacy of drugs targeting these microenvironmental cell types. Because these cells
- 6) The authors refer to a concept of vascular supply and how tumor cells may be proximal or distal to vasculature-like structures. It is not exactly clear what the significance of vascular supply is when they have not demonstrated that there are contiguous endothelial layers that conduct any fluid.
- 7) It is not clear what the authors suggest when they draw relationships between more macrophages and less supply in the middle of the TMO?
- 8) It is not clear why there is a need to assess the Observer score –Why is this an issue? AI tools should provide a quantitative assessment of remove observer artifacts.
- 9) MDA-MB231 cells are mesenchymal and do not have any obvious epithelial properties; the reasoning for expressing EMT-promoting transcription factors does not make any sense. Hence, it is unclear if the data presented in Figure 8 can be interpreted with any significance.
- 10) A minor point is that growing established cell lines in 3D culture is not referred to as organoids, which are reserved for 3D cultures derived primary tissue or tumor-derived cells or stem cells.

Response to the reviewers' comments

First of all, we would like to thank the reviewers for taking the time to read our manuscript and for all their positive and constructive comments, which, in our opinion, helped us a lot to improve the work. Below, we discuss the comments point-by-point and explain how the reviewers' suggestions have been incorporated into the manuscript. Inspired by the reviewers' comments, we also conducted additional experiments in mouse models to validate our screening results. To make it easier for the reviewers to follow the changes we made in the manuscript, we highlighted in yellow the passages that have been rewritten or added since our first submission.

In particular, we would like to thank the reviewers for pointing out what was missing in explaining the motivation behind our study. Thus, we now put extra effort into emphasizing the intentions behind our project to make our motivation entirely clear: The objective was to develop a platform that facilitates the efficient side-by-side evaluation of drug candidates during pre-clinical drug development and not to propose an alternative approach to personalized medicine. These two objectives are very different and result in very different challenges. After extensive rewriting of the introduction and discussion, we believe our approach is now better defined and our intentions easier to follow. (We elaborate a bit more on the specific and distinct challenges in establishing a TMO screening setting for personalized medicine in our response to reviewer #4's comment 4 below.)

The specific demands resulting from developing a screening platform for drug candidates also lead to our deliberate decision to use established cell lines and primary cells, but not patient material. That decision was neither a limitation nor a cheap compromise. (We have access to patient material and are in the middle of a project establishing TMOs from it.) Actually, it is an advantage and strength because it allows us to generate spheroids with a highly complex TME architecture with high reliability and reproducibility. We are confident we demonstrate and now highlight this high reproducibility in our manuscript. This high reproducibility is fundamental for being able to compare drug effects, and it is not achievable with the inherent variability of patient-derived material.

Moreover, it has to be mentioned that the published work on establishing patient-derived organoid cultures rarely aimed at recreating the TME. We are not aware of any publication that demonstrates the recreation of a TME nearly as complex as shown in our manuscript. For example, using patient-derived material alone, it is not possible to recreate a (pseudo)vascular network. Endothelial cells, but also essentially all immune cells, cannot be functionally cultured even in the context of a 3D organoid. This is our own experience with patient material, and it can also be extracted from the literature. Even the tumor cells in patient samples cannot be easily cultivated as organoids. Steps have to be taken to prevent the overgrowth by fibroblasts.

We have highlighted these differences in detail and are confident that the reviewers will find that the revised manuscript effectively conveys our intentions and the novelty of our work to the reader.

Thank you again for your time, and your valuable input!

Comments to specific remarks by the reviewers:

Reviewer #1 (Remarks to the Author):

This manuscript developed a platform to generate tumor spheroids with different cell types and then implemented light sheet fluorescence microscopy (LSFM) to image and semi-quantitative analyze the spheroids. This combined platform/strategy provided many advantages over homogenous spheroids and other evaluation approaches. The authors further implemented the platform to screen and evaluate various antifibrotic and antiangiogenic drugs. The methods and approaches are innovative. But there are some concerns related to the verification of this platform, the similarity of generated spheroid model to the native tumor, and overall applicability for screen the drug for clinical usage.

1. The authors need to prove the accuracy of LSFM based analysis method. This is very important. The ratio of tumor:fibroblast was initiated from 1:20. The analysis of LSFM on day 0 should be presented.

The reviewer probably suggests evaluating LSFM images of TMOs at day 0. Unfortunately, this is technically not possible. On day 0 of the TMO generation, only a loose suspension of the cells on top of the agarose coating exists. This suspension cannot be fixed and stained for LSFM imaging. It takes 48-72 h until solid TMOs are formed. Furthermore, it is not clear to us in which way the cell suspension shall be different from the sum of well-defined and counted cells we just seeded on top of the agarose.

2. How comparable is the generated TMO to native breast tumor tissues? The use of normal fibroblasts and THP-1 raises concerns

The tumor cell lines we used to generate the TMOs were established in the 1970s and 1980s. Therefore, a direct and meaningful side-by-side comparison of the TMOs to native tumors is not possible.

In the manuscript, we describe observed differences in TMOs generated from different breast cancer lines that mimic histological differences in human breast cancer sections. E.g., even dispersion of invasive breast cancer cells in a stroma matrix vs separation of tumor cell islands from stromal strands. Or, on the other hand, embedding and shielding of microvessels in the stromal strands vs direct contact of microvessels with tumor cells.

Fortunately, we can be less concerned with the use of normal fibroblasts. These cells have been shown to be very responsive to their environment. We are sure that these cells, after the prolonged co-cultivation, are no longer “normal” fibroblasts. Using tumor-associated fibroblasts (TAFs) would be a possibility. TAFs are easy to isolate and handle. However, using fresh patient-derived TAFs in every screening experiment would introduce another element of variance, as they would differ from patient to patient. This would preclude inter-assay comparison of results. Establishing TAFs in cell culture, which could be

used in all subsequent assays, would mean accumulating culture artifacts and losing any benefits over NHDFs. Thus, we think the use of NHDFs is more than a good compromise and fully justified as it also mimics the stromal infiltration into real tumors and the subsequent transformative adaptation of the fibroblasts to the new tumor environment.

THP-1 cells are indeed a compromise as these cells are transformed and not real monocytes. They have been shown to be able to acquire characteristics of either M1 or M2 macrophages, depending on context and stimulation. In the absence of cultivatable monocytes, we regard them as a necessary but viable compromise. Our results, presented in the manuscript, indeed show that these cells are a satisfactory, although maybe not a perfect, stand-in for monocytes.

3. What medium was used for the culture of TMO? How the medium affect the TMO component changes.

We used EGM-2 (Lonza). The information is now available more prominently in the manuscript.

Initially, we performed some experiments in DMEM (10%FBS) with results that needed improvement. In particular, the formation of a PV network was strongly impaired. The results in EGM-2, on the other hand, were, from the beginning, good and consistent, so we continued to use this medium.

4. One of the major concerns is that the whole analysis was based on LSFM. There was no other method was used to verify the results. For example, in Fig. 6, some other cellular/molecular assays should be used to verify the concept of vascular effect index.

We decided to verify the results and the concept of a VEI by re-testing several of the drugs in a murine breast cancer model. Short of going into patients, we are confident this is the best way to validate the relevance of our *in vitro* testing system.

The *in vivo* data is shown in Extended data figure S8c-g. The reviewer will find that the *in vivo* experiments not only verified the general effect of the tested drugs but also that the VEI-based ranking of the drugs in the mouse model is consistent with the ranking in the *in vitro* TMO system.

5. There are many limitations and restrictions of the approach and platform. The authors should thoroughly discuss these points.

We comprehensively revised the discussion in order to address the reviewer's recommendations. This includes an expanded discussion of the challenges ahead to overcome the limitations of the system

Minor:

1. The term "humanized" is not suitable to be used in this study.

The reviewer is, of course, correct. We changed and clarified this throughout the manuscript.

2. The term "microorganoid" is also not accurate in this study.

We understand the reviewer's concern as, in the literature, the terminology is tricky and often inconsistent. Lancaster and Knoblich (Science, 2014) defined organoids as follows:

"Organoids are derived from pluripotent stem cells or isolated organ progenitors that differentiate to form an organlike tissue exhibiting multiple cell types that self-organize to form a structure not unlike the organ *in vivo*."

(Other definitions, in general, use the same hallmarks to define organoids).

For sure, our spheroidal structures are not derived from stem or progenitor cells (let aside the progenitor-like features of the utilized cancer cells). However, they are self-organizing, exhibit multiple cell types, and form structures not unlike tumors *in vivo*.

Moreover, tumors, in contrast to other organs, do not arise from stem cells. The cells of the TME are mainly recruited or coopted from fully differentiated cells of the host tissue. Tumor blood vessels develop mainly by sprouting angiogenesis, although tumors might activate endothelial progenitor reservoirs under certain circumstances like therapy-induced acute hypoxic stress (e.g. Shaked, Henke, et al. Cancer Cell, 2008). Tumor-associated fibroblasts are derived from fibroblasts of the surrounding tissue (Fotsitzoudis, Cancers, 2022) and macrophages from infiltrating monocytes of the circulation. Thus, the way our spheroidal structures self-assemble from tumor cells and various differentiated cells resembles, to a certain degree, tumorigenesis. Maybe the best comparison is the establishment of a novel tumor after metastatic seeding at a distal site. Considering this, and keeping with the "tumor as an aberrant organ" concept (Jain, Cancer Res., 2004), we do not think naming our tumor-like structures "tumor organoids" is misleading.

We would like to provide the readers with a clear term that conveys the complexity of the spheroidal structures they are to encounter in our manuscript. It makes it also possible to immediately distinguish these structures from the much simpler multicellular tumor spheroids we also used in several experiments. This is why we would prefer to keep the term as it is.

Reviewer #2 (Remarks to the Author):

I co-reviewed this manuscript with one of the reviewers who provided the listed reports. This is part of the Nature Communications initiative to facilitate training in peer review and to provide appropriate recognition for Early Career Researchers who co-review manuscripts

Thank you very much for your time and your input on our research! So far, we were not aware of this program by NatComm. We think it is a great idea. All the best for your future career!

Reviewer #3 (Remarks to the Author):

The authors present a platform for screening tumor organoids and demonstrate its utility with several cases studies.

The topic is of high importance, and the study seems to be thorough. However, after reading the manuscript several times, it remains unclear to me where exactly is the novelty of the results and what is the take-home message for readers. Perhaps it is due to the way results are structured and presented, but it remains vague. Is the novelty in the way of growing the specific micro-organoids? Is it in data analysis? Or in the findings of the case studies?

Thank you very much. We take this concern very seriously. We revised several passages in the manuscript to make the novelty better understandable for all readers. In particular, we rewrote large parts of the introduction in response to the reviewer's concern. Following, we will elaborate a bit on what we think is new in our work, as we can here describe in more detail what has to be presented more condensed in the limited space of the manuscript. We believe the following thoughts are now also properly featured in the revised manuscript's introduction and beyond.

It is possible that the impression of vagueness the reviewer experiences stems from the fact that we consider several aspects of our work as novel and previously did not highlight one aspect in particular. We do not think that this is surprising: If our work would only focus on improving a single part of the challenge to develop a platform for the testing of TME-targeted drugs, it would be more suitable for a more specialized journal than Nature Communications.

As we see it, central novelties are:

1. The generated TMOs

We are not aware of any publication describing the recreation of a similarly complex tumor microenvironment (pseudovascular network, fibrotic matrix, embedded macrophages) at such a small, high-throughput-enabling scale. Much less, with such simple methods.

2. The utilization of LSFM for recording drug effects

Using state-of-the-art microscopy enables the simultaneous mapping of a multitude of parameters. In their entirety, these parameters capture drug effects in a novel comprehensiveness. We are not aware that anyone has used 3D microscopy in a similar way to assess drug effects in an *in vitro* screening system.

3. The ranking method

We developed a new algorithm that enables the ranking of drugs according to different effects that might be desirable.

In addition, we are sure that readers from the biotech sector will appreciate the robustness, reliability, versatility, and modularity of our approach, which is also, in our opinion, unparalleled.

The manuscript's central topic is the new platform, but it is not clear how an interested reader can reproduce or implement the platform as a tool. The concrete actionable details to reproduce the platform seem to be missing.

Thank you for pointing out that this was unclear. In the revised manuscript, we ensure that all methods are described in greater detail. We believe that a typical graduate student with access to standard cell culture equipment and a standard-equipped microscopy core facility should be able to reproduce our work easily, as the strength of the demonstrated platform is its robustness and simplicity. Here are some details:

Generation of the TMOs

This can be done in any lab with standard cell culture equipment. The method, which we describe in detail, is so simple that it should even be possible to reproduce by an undergraduate student with a minimum of cell culture experience. In fact, one of our co-authors (Daniel Szi-Marton) was an undergrad when he performed the whole routine of culturing and treating the TMOs, staining and imaging them, and processing the image files.

Imaging of the TMOs:

This can be done at any research institution where access to a scanning microscope with 3D capacity is available. If an LSM microscope is not available, a confocal microscope will do (with the possible compromises in image quality, as discussed below).

Image processing and evaluation

Although we used a convenient, rather expensive software package (Imaris), there are also free-of-charge solutions available (e.g., NIH's 3D Slicer, <https://www.slicer.org/>). The costs for Imapris or similarly powerful commercial software are usually not prohibitive for core facilities at major research institutions.

Since I personally specialize in microscopy, below are my specific questions about this part.

Major

Methods, Image Acquisition, lacks many details that are expected:

- for adequate description of a custom microscope, a CAD model and sufficiently detailed technical drawings (2D and 3D, including the light paths, lenses, and general optomechanical arrangement) should be presented for readers to grasp the design principles used, if not to fully reproduce the design.

That is a great suggestion, and we have added a 3D drawing of the microscope's setup (Extended data figure S11b). We are confident that together with the detailed description of the individual components provided in the M&M section, interested readers can now easily comprehend the design.

- I. 960-961, "custom-built light sheet microscope, with a 20x objective," - which objective? Manufacturer, model, NA, immersion type? I guess this is Leica HCX APO L 20x/0.95 IMM, introduced in 1979, but it should be stated more clearly. I would either remove "with a 20x objective" altogether from the first sentence, or specify it fully.

Yes, it is the Leica HCX described further down in the paragraph. We have now specified it fully in the first sentence (now l. 1034-35).

- "The hydrogel plugs were attached to the head-mounted xyz stage using a pincer clip and submerged in the EtCi-filled imaging chamber." A photo or a drawing would be very helpful to visualize this.

We thank the reviewer for bringing this up. A photo and a drawing of the setup in the imaging chamber have been added as Extended Data Figure S11a.

- l.964 "A custom-built LSFM setup tailor-made for organ imaging" Is it another setup, or the same one?

Yes, the reviewer is, of course, correct. We changed the sentence to make it better understandable. We apologize for the unclear wording.

- what software was used to control the custom microscope(s)? Is this software already published? If not, do authors plan to publish it? If not published along with microscope design, it hardly counts as an adequate description of a platform.

We have to apologize for forgetting to put this information into the manuscript. It was only included in the NPG reporting summary, but by mistake, it was not added to the manuscript. The reviewer will now find the information (Andor iQ2 (Oxford instruments Abington UK, formerly Andor Technologies)) in the methods section.

Minor:

- MCTS abbreviation is introduced in line 156, but not explained until l. 837

Thank you for pointing out this mistake. We included the definition in line 156.

- I would avoid double line spacing, it makes the text stretched and difficult to read.

We understand that the double-line spacing makes it especially tedious to print out the stretched text for review. However, this is the standard spacing demanded by virtually all journals that give strict formatting guidelines.

- ethyl cinnamate is usually abbreviated as ECi, not EtCi

Thank you! We changed this. The reviewer is, of course, correct that the vast majority of authors prefer ECi.

- antibodies dilution in Methods must be stated.

Absolutely correct. We have included a table with the AB dilutions (supplemental table 5)

Clarity/presentation

"To guarantee undistorted 3D imaging, we decided to use light sheet fluorescence microscopy". What exactly is undistorted imaging? Why does LSFM provide such "undistorted" imaging?

We fully agree with the reviewer that our wording was unclear. Our microorganoid samples are up to 500 μm in size. Imaging samples of such size by conventional confocal laser scanning microscopy (CLSM) can cause imaging artifacts due to (i) photobleaching (in CLSM the excitation path is colinear to the detection path) and (ii) slower acquisition times (point detector vs camera). Hence, we used LSFM as the imaging method of our choice. In LSFM, only the imaging plane is excited, and the fluorescence light is collected by a camera. Thus, 3D images can be acquired faster, which causes fewer artifacts. We clarify that by changing the main text (lines 200-202) as follows:

Due to the size of the TMOs (~300-500 μm in diameter), we decided to use LSFM, which allows fast 3D imaging of large fields of view while minimizing photobleaching....

The text is often difficult to follow. For example, in the Discussion I.721: "Thus, throughput is limited by investment in computational power only. An exception is LSFM, where no commercial solution for automated sample throughput is available." I guess authors are trying to say that computational analysis is easily scalable (is it?), but high-throughput imaging is challenging, and light-sheet microscopy helps addressing this bottleneck. But the sentence is written ambiguously, and this is just one example.

We changed the mentioned section and made it much clearer now.

In addition, we let one of our students read the entire manuscript and mark all sections she found not easily understandable. After the subsequent corrections, we are confident that the text is now significantly improved and easier to comprehend.

To better structure the text, especially Introduction and Discussion, I recommend Mensh B, Kording K (2017) Ten simple rules for structuring papers. PLoS Comput Biol 13(9): e1005619. <https://doi.org/10.1371/journal.pcbi.1005619>

Thank you for recommending this great article.

Reviewer #4 (Remarks to the Author):

In this manuscript, the author presents results from an effort to promote the assembly of multiple cell types, including tumor cells, macrophages, endothelial, and fibroblasts, to form tumor microorganoids (TMO). They report a methodical microscopic analysis of the temporal steps in the formation of TMO and interesting computation renditions of the process. They propose that the use of established cell lines for TMO makes the system more robust and reduces the risks associated with using primary tumor materials. The authors also demonstrate the utility of the model to assess the effectiveness of therapies aimed at targeting cells in the microenvironment. Although the study uses innovative imaging approaches, it lacks conceptual rigor and is too early to make impactful conclusions.

1) The authors argue that the failure of drugs in clinics is due to a lack of good models and claim that the TMO platform was developed to address the challenge, yet the studies presented are using systems that are unproven to model patient tumors, let alone addressing the challenge they propose to address.

We apologize for this misunderstanding. We definitely argue – and we do not consider this to be controversial – that better pre-clinical models would help to reduce failures in the clinic. Considering our work, we are simply confident that we were able to demonstrate that the platform we present is able to provide the user with an unprecedented range of information about a drug's effects in a complex microenvironment. That is a major challenge in creating better models, and we addressed it.

Our TMOs definitely model patient tumors – in some respect. In others, certainly not. (“A model has to be wrong in some respect. Otherwise, it is not a model but the thing itself.”) Again, we do not find it controversial at all to claim that our TMOs model patient tumors much better than simple 2D cultures of tumor cells or standard 3D MCTS. In some respects, they represent the situation in patients much better than most patient-derived organoids. They model TC-fibroblast-, TC-endothelial, and TC-macrophage interaction. They recreate a pseudo-vasculature. They can be used to test how all these interactions and components react to drugs. Patient-derived organoids fail in all these aspects. On the other hand, of course, patient-derived organoids can give oncologists important information on how the tumor of a particular patient might react to a specific treatment. That is information our system cannot deliver, and it was never intended to be able to.

Taking all this into account, we hope that the reviewer generally appreciates that we indeed successfully addressed major challenges in developing a drug testing platform that enables the pre-clinical testing of drug candidates in a more realistic setting.

2) Whether the platform has any potential for modeling clinical response is unclear. Most of the drugs used in the study have failed in breast cancer clinical trials, raising concerns about the utility of the platform.

We are surprised by this statement as five of the ten substances we tested have not even been assigned a tradename – something most companies do when drug candidates are selected for clinical development. In fact, we show in the list below that the drugs we tested **have not failed in breast cancer clinical trials**.

Moreover, a major incentive in developing a complex assay system for drugs that target the TME was to provide more detailed knowledge about the actual effects of a drug candidate before entering clinical trials. This knowledge can guide the design of clinical trials. It is reasonable to assume that many drugs fail trials due to ill-informed setups (false patient stratification, wrong sequence of drug application in combination therapy, etc.) as a consequence of a lack of knowledge. Thus, the re-testing of candidates that previously underperformed in trials (again, what we did not do!) would **not** render the study useless.

We fact-checked the claim that the drugs we tested **failed in breast cancer clinical trials** by consulting Pubmed and clinicaltrials.gov:

- Axitinib is approved for the treatment of RCC. A single (!) phase II trial (NCT00076024) of axitinib has been performed in breast cancer (Docetaxel +/- AXI). In this trial, OR was significantly increased, but overall results were not encouraging enough to continue with additional trials.
- Beta-APN: is not a substance anyone would send into costly clinical trials. Not surprisingly, we were not able to find a single trial involving beta-aminopropionitrile for the treatment of breast cancer or any other neoplasms.
- Minoxidil: we were not able to find a concluded trial involving minoxidil for the treatment of breast cancer or other neoplasms. NCT05272462 (“Oral Minoxidil for the Treatment of Recurrent Platinum Resistant Epithelial Ovarian Cancer”) is still recruiting.
- Pomalidomide is approved for the treatment of multiple myeloma and Kaposi sarcoma and has been tested in a few other malignancies. We were not able to find a single trial involving pomalidomide for the treatment of breast cancer.
- Pirfenidone has been tested in cancer patients, but to treat radiation therapy-induced fibrosis – not the existing tumors. NCT05704166 (PirfenidoneVsPlacebo as Prophylaxis Against Acute Radiation-induced Lung Injury Following HFRT in Breast Cancer Patients) is still recruiting. We were not able to find a trial involving pirfenidone for the treatment of breast cancer.
- RS-504393: We were not able to find a single trial involving RS-504393 for the treatment of any cancer.
- Deshydroxy LY-411575: We were not able to find a single trial involving Deshydroxy LY-411575 for the treatment of any cancer.
- UK-383,367: We were not able to find a single trial involving UK-383,367 for the treatment of any cancer.
- ZD-7155: We were not able to find a single trial involving ZD-7155 for the treatment of any cancer.
- LDC1267: We were not able to find a single trial involving LDC1267 for the treatment of any cancer.

In addition to the above-indicated major concern, there are multiple concerns:

3) The computational pseudo-vascular rendition images, while providing nice images, seem to have unclear biological significance as the authors do not demonstrate that the endothelia form a contiguous tubular structure that is capable of conducting fluids.

We elaborate further down in our reply to the reviewer's comment #6 on this topic. In brief:

1. The pseudovasculature in the TMOs is, of course, not matured enough to conduct fluids. Such a maturation step is not possible without a connection to actual fluid flow.
2. The pseudovasculature replicates other aspects of the microvascular network in tumors.
3. Connecting the pseudovessels to a fluid-transporting system (e.g., by implantation in a CAM assay) would most likely result in a certain degree of fluid transport throughout the PV network and maturation of the pseudovessels. It would be surprising if not, as it has been done numerous times.
4. Aside from the futility of performing a highly redundant experiment, a connection to a fluid delivery system would destroy all benefits of our screening system (an entirely human setup, robustness, throughput, parallel screening, and ranking of drug candidates).
5. Neither a successful clamping of the TMOs PV to a fluid delivery system nor the – highly unlikely – failure of such an effort would affect any claims we make in the manuscript.

4) The authors have not supported the reasoning to believe the observed organization models anything relevant to breast cancers in patients. Given there is a great deal of patient-to-patient in the molecular and morphological organization of the tumors, it is not clear what the authors are modeling.

As outlined above, we were not aiming to re-create the TMEs of specific tumors directly derived from patients in an attempt at personalized medicine. The objective was to create a platform for the reliable evaluation of drug candidates in a pre-clinical setting. Consequently, the main focus was on a robust, highly reproducible setup. This explains our utilization of established, well-characterized tumor cell lines. The cell lines were established decades ago, and in part not from primary tumors but from pleural effusions. To expect that these tumor cell lines recreate a specific TME (which one should be held in comparison?) is not plausible.

We show at length that the individual cell lines are creating their own specific microenvironment, influencing macrophage proliferation and incorporation, stroma strand formation, and architecture and size of pseudo-vascular networks. This broad diversity of the TME architecture in our TMOs *reflects* aspects also found in the histology of breast carcinomas.

The reviewer correctly mentions the “**patient-to-patient [diversity] in the molecular and morphological organization of the tumors**”. Following this important observation is the fact that it is not advisable to aim to recreate the molecular and morphological organization of particular tumors. Again, which

one of them? The whole range of possible breast tumors? The morphological diversity of breast carcinomas is so wide that it can be said that each tumor is unique, as the reviewer also implies with the term “patient-to-patient”. Moreover, the diverse histological characteristics are not strongly linked to molecular subtypes that infer response to certain treatment modalities, to the stage, or to the grade of breast tumors. Thus, the only setting where it makes sense to aim to recreate the characteristics of a particular tumor is in personalized medicine, thus, in organoid cultures from specific patients.

Two years ago, we started a project that aims to recreate the TME of specific HNSCC tumors in the form of microorganoids. In this project, we have fresh tumor material at hand and also have the possibility to process the original tissue for histological assessment. *In theory*, this approach will make it possible to recreate a specific TME and compare it to the tumor of origin. However, in *praxis*, there are huge challenges. TMOs generated from patient material are not self-organizing. Steps must be taken to prevent overgrowth by TAFs that even outgrow tumor cells. All other cells of the TME (ECs, immune cells) cannot be cultured from patient material. They have to be augmented from other cultures. So far, we made progress but have not yet been successful in creating organoids from fresh material as complex as the one we created from cell lines for the manuscript at hand. Most importantly, even after two years into this project, we struggle to define the hallmarks by which we can measure the degree to which we were able to truthfully recreate the original tumor.

5) It is unclear if the co-culture impacts the molecular state of any cell types when they are part of the TMO? If being part of the TMO does not significantly alter the cell phenotype compared to the cells grown alone, then one may wonder if one needs to generate TMO to assess the efficacy of drugs targeting these microenvironmental cell types. Because these cells

In the manuscript, we provide evidence that this is no longer a concern. We show in the paper:

1. Molecular changes in tumor cells within several TMO models, with respect to the markers CD44 and ALDH1A1 (Fig. 7i,j Extended Data Figure S9c,d). These changes are induced by 3D vs 2D cultivation but are also differentially influenced by the addition of the various cell types that were included in the TMOs.
2. We demonstrate that the various cell types in the TMOs influence the response to CTX of the tumor cells. This is a phenotypic difference, and it is not plausible that this occurs without molecular changes in the tumor cells.
3. Most of the manuscript details changes in the PV architecture, the variations in fibroblast distribution, or macrophage agglomeration. These are all phenotypic changes. Moreover, these phenotypic changes also occur in response to confrontation with different tumor cell lines within the TMO environment.

The reviewer claims that “**If being part of the TMO does not significantly alter the cell phenotype compared to the cells grown alone, then one may wonder if one needs to generate TMO...**”.

Yes, TMOs are an important tool for the following reasons:

1. Endothelial cells grown alone do not form pseudovascular structures. So yes, their phenotype is clearly altered. How would the reviewer assess vascular disrupting properties on endothelial cells grown alone?
2. We show that chemotherapy affects tumor cells differently, not only when grown in 3D compared to 2D but also when co-cultivated with other cells. So yes, being part of the TMO alters their phenotype.

6) The authors refer to a concept of vascular supply and how tumor cells may be proximal or distal to vasculature-like structures. It is not exactly clear what the significance of vascular supply is when they have not demonstrated that there are contiguous endothelial layers that conduct any fluid.

Of course, the pseudovasculature in our TMOs is not connected to a circulation.

Our organoid-like structures contain endothelial networks that are void of fluid flow. Therefore, we refer to it as a pseudo-vasculature, not as a vasculature. We do not see this pseudo-vasculature as an entryway for drugs and do not claim that it is, but we have clarified this more thoroughly in the text. The pseudo-vasculature is a model for the microvasculature networks observed in tumors. And we demonstrate that drugs affect this pseudo-vascular network. Using a breast cancer mouse model, we were now able to show that the effects seen on the pseudo-vasculature in the TMOs align well with the effects on the tumor vasculature in mice (Extended Data Figure S8). The vessels in the murine tumors are, of course, perfused. This underlines our claim that the pseudo-vasculature in the TMOs is a valid model for the tumor microvasculature.

In a 96-well *in vitro* format, engineering a vascular network that conducts fluids is not possible. Thus, to accommodate the wishes of the reviewer to see the fluid flow, we would have to establish a completely different system, e.g., by implanting our organoid-like structures into a microfluidic circuit, embryonic chorioallantoic membrane (CAM) of chicken eggs or murine ischemic hindlimbs. These experiments have been done numerous times before (we ourselves conducted CAM implantation), and uniformly, the implanted pre-vascularized structures anastomose with capillaries of the recipient. It would be a surprise if our organ-like structures would behave differently. Moreover, conducting such a transfer experiment would mean establishing a completely different system. This system would be void of the beneficial characteristics (robustness, high reproducibility) of our initial system, and drug screening would be impossible (or severely impeded). Most importantly, no results seen in this new system could be transferred back to our original setup.

When we refer to the concept of vascular supply, we do not claim that the pseudo-vasculature has a supplying function. This is in reference to the pseudo-vasculature being an analogous model for the tumor vasculature. When we see that a treatment reduces the density of the pseudo-vasculature in the TMOs, it should also reasonably reduce microvessel density in a tumor. This, again, should reduce supply in the surrounding tumor volume. We went through the text of the manuscript and clarified this (lines. 275-276).

7) It is not clear what the authors suggest when they draw relationships between more macrophages and less supply in the middle of the TMO?

It was not our intention to suggest such a relationship. We carefully reviewed the manuscript to make sure that no such misleading statement is in the text.

8) It is not clear why there is a need to assess the Observer score –Why is this an issue? AI tools should provide a quantitative assessment of remove observer artifacts.

We agree with the reviewer that moving the assessment to a machine-learning environment will be a logical next step. However, “AI tools” are not magic boxes that can return answers to arbitrary questions after being fed novel data. Machine learning/deep learning routines have to be trained. For training, annotated data sets are necessary. Just as an example: to train facial recognition software, huge data sets with images annotated by humans (= observers) are necessary to define photos as “showing the same person” or “showing a different person”.

In letting observers score (= annotate) our 3D image stacks, we took the first step to ML/DL-based assessment. So far, our accumulated data (close to 200 TMOs) is not yet sufficiently large to start training an ML routine. In addition, such an ML tool has to be created. To our knowledge, there are no “off-the-shelf” ML solutions for the classification of 3D image stacks. However, that is definitely something we are looking into and looking forward to doing. We address this subject in our discussion.

BTW, as “AI tools” are trained on data sets annotated by humans, they are unable to remove “observer artifacts”. They might even amplify them. This was demonstrated by the biases displayed by NLP models or face recognition software (<https://www.theverge.com/2016/3/24/11297050/tay-microsoft-chatbot-racist>, <https://sitn.hms.harvard.edu/flash/2020/racial-discrimination-in-face-recognition-technology/>).

9) MDA-MB231 cells are mesenchymal and do not have any obvious epithelial properties; the reasoning for expressing EMT-promoting transcription factors does not make any sense. Hence, it is unclear if the data presented in Figure 8 can be interpreted with any significance.

We have to respectfully disagree here. MDA-MB-231 is not a mesenchymal cell line. It is a mammary adenocarcinoma cell line and, thus, of epithelial origin. This origin has, to our knowledge, never been questioned. Expression of epithelial markers in MDA-MB-231 cells has been demonstrated numerous times (Gonzalez-King, Cancer Gene Ther. (2022); Qin, JExpClinCanRes (2018); Pan, SciRep (2016); Liu, SciRep (2019); etc., etc.). Consequently, ATCC lists MDA-MB-231 as an epithelial cell line with epithelial morphology (<https://www.atcc.org/products/htb-26>).

For sure, MDA-MB-231 cells also express mesenchymal markers and show the effects of EMT (invasiveness, loss of polarization, etc.). Of the breast cancer lines we used in our project, MDA-MB-231 is, without doubt, the one that shows

the most pronounced mesenchymal characteristics. However, this does not render them a mesenchymal cell line. EMT in cancer, other than in early development, is not understood as a full commitment to a mesenchymal fate. It is the acquisition of characteristics and molecular markers usually found in cells of mesenchymal origin. Moreover, it is understood to be dynamic and dependent on cues from the environment (see EMT <-> MET switches).

Basal expression levels of Snai1 and Twist1 are low in MDA-MB-231. Overexpression, as we demonstrate (Extended Data Figure S10b), increases these levels significantly. Thus, it can be reasonably assumed that the OE of Snai1 or Twist1 will result in considerable changes in the cell line's molecular characteristics and in their effect on the surrounding environment – which we clearly demonstrate in our experiments.

10) A minor point is that growing established cell lines in 3D culture is not referred to as organoids, which are reserved for 3D cultures derived primary tissue or tumor-derived cells or stem cells.

We understand the reviewer's concern as, in the literature, the terminology is tricky and often inconsistent. Lancaster and Knoblich (Science, 2014) defined organoids as follows:

“Organoids are derived from pluripotent stem cells or isolated organ progenitors that differentiate to form an organlike tissue exhibiting multiple cell types that self-organize to form a structure not unlike the organ *in vivo*.”

(Other definitions, in general, use the same hallmarks to define organoids).

For sure, our spheroidal structures are not derived from stem or progenitor cells (let aside the progenitor-like features of the utilized cancer cells). However, they are self-organizing, exhibit multiple cell types, and form structures not unlike tumors *in vivo*.

Moreover, tumors, in contrast to other organs, do not arise from stem cells. The cells of the TME are mainly recruited or coopted from fully differentiated cells of the host tissue. Tumor blood vessels develop mainly by sprouting angiogenesis, although tumors might activate endothelial progenitor reservoirs under certain circumstances like therapy-induced acute hypoxic stress (e.g. Shaked, Henke, et al. Cancer Cell, 2008). Tumor-associated fibroblasts are derived from fibroblasts of the surrounding tissue (Fotsitzoudis, Cancers, 2022) and macrophages from infiltrating monocytes of the circulation. Thus, the way our spheroidal structures self-assemble from tumor cells and various differentiated cells resembles, to a certain degree, tumorigenesis. Maybe the best comparison is the establishment of a novel tumor after metastatic seeding at a distal site. Considering this, and keeping with the “tumor as an aberrant organ” concept (Jain, Cancer Res., 2004), we do not think naming our tumor-like structures “tumor organoids” is misleading.

We would like to provide the readers with a clear term that conveys the complexity of the spheroidal structures they are to encounter in our manuscript. It makes it also possible to immediately distinguish these structures from the

much simpler multicellular tumor spheroids we also used in several experiments. This is why we would prefer to keep the term as it is.

Reviewers' Comments:

Reviewer #1:

Remarks to the Author:

The authors addressed most of the questions and the manuscript was significantly improved. Although the authors did not compare the TMO with native tumor tissues, the technique and approaches are innovative and promising.

Reviewer #2:

Remarks to the Author:

Reviewer #3:

Remarks to the Author:

The authors answered all my major concerns and significantly improved the clarity of presentation. They also provided essential details for the microscopy part. I have found only several minor issues in the revised manuscript:

The new Extended Data Figure S11 is really helpful for understanding the imaging methodology of the work, and nicely done. However, this 3D design reminds me of two other recent papers that used knife-edge mirror for similar purposes:

1. RM Power, A Schlaeppli, J Huisken. Compact, high-speed multi-directional selective plane illumination microscopy. *Biomedical Optics Express*, 2023.

2. N Vladimirov, F Preusser, J Wisniewski, Z Yaniv, RA Desai, A Woehler, S Preibisch. Dual-view light-sheet imaging through a tilted glass interface using a deformable mirror. *Biomedical Optics Express*, 2021.

It would be nice if authors cited key methodological papers related to their work, including microscope development (whether they derived their design from those papers or not).

- which XYZ stage was used for sample motion?

- the AHF Analysentechnik is a dealer, the manufacturer of the mentioned filters is Semrock.

- is the TMO (individual organoid body) segmentation done in Imaris manually, or in a automated/scripted way?

- related, is the cell/vasculature segmentation in Imaris scripted, or done manually? If manually, this is hard to claim good scalability and possibility for automation. If scripted (batch processing), the code should be published along with the manuscript, and be available for review.

Reviewer #4:

Remarks to the Author:

In this revised manuscript, the authors have taken it upon themselves to argue that they were correct in the first place, and the concerns raised are either not real concerns or irrelevant to their conclusions.

The concerns I raised were aimed at giving an opportunity for the authors to make their manuscript more scholarly and balanced by acknowledging/discussing caveats and limiting the

over-interpretation of their findings. However, it is disappointing that the authors are unable to see the risk their over-interpretation poses and are also unable to recognize the caveats of their studies in a scholarly manner. I am choosing not to provide a detailed response because the authors have not addressed my original concerns. I will, however, provide a couple of examples to explain myself and will leave it to the journal and authors to associate themselves with the impact (positive or negative) this manuscript will have in the field.

The authors are unable to understand the difference between cancer cells with epithelial or mesenchymal properties and the cell type the cancer originates from. The end cell phenotype does not need to be identical to the originating cell phenotype. An epithelial cell-derived tumor may lose all its epithelial properties during its genesis and become a mesenchymal cell. So, of course, no one will question the origin, but they will know what the end phenotype is. When I indicated MDA-MB231 are mesenchymal in nature and do not have any classical epithelial properties (e.g., e-cadherin expression and cobblestone morphology), I was not referring to their originating cell type. But the authors chose to respond with a lecture on how it is a mammary adenocarcinoma, etc. etc. To me, their response highlights their dogmatism more than anything else.

The minor concern about the organoid nomenclature of what is considered in the field to be a norm was to encourage them to consider their caveats and de-risk themselves from being received by readers as scientists who do not recognize the norm for the field. In response, the authors provide a response that is largely unscholarly.

Lastly, when I highlighted most of the drugs that have failed clinical testing in breast cancer, it was to give the authors a chance to demonstrate that their TMO platform is better than the existing platform for the validation of candidate drugs either by comparing other platforms or choosing drugs that did or did not succeed in the clinic (the end goal they are trying to model in TMO) and demonstrate that drugs that work in clinic work in their TMO model and those that did not work do not work. Taking drugs that were never tested in the clinic and show efficacy or lack thereof is a drug discovery effort and not a platform validation effort.

Response to the reviewers' comments

We would like to thank once again the reviewers for taking the time to read our revised manuscript and for all their positive and constructive comments. Your input helped us tremendously in improving our manuscript.

Reviewer #1 (Remarks to the Author):

The authors addressed most of the questions, and the manuscript was significantly improved. Although the authors did not compare the TMO with native tumor tissues, the technique and approaches are innovative and promising.

Thank you again for your time and effort in evaluating our manuscript.

Reviewer #2 (Remarks to the Author):

Thank you for your input!

Reviewer #3 (Remarks to the Author):

The authors answered all my major concerns and significantly improved the clarity of presentation. They also provided essential details for the microscopy part. I have found only several minor issues in the revised manuscript:

The new Extended Data Figure S11 is really helpful for understanding the imaging methodology of the work, and nicely done. However, this 3D design reminds me of two other recent papers that used knife-edge mirror for similar purposes:

1. RM Power, A Schlaeppli, J Huisken. Compact, high-speed multi-directional selective plane illumination microscopy. *Biomedical Optics Express*, 2023.
2. N Vladimirov, F Preusser, J Wisniewski, Z Yaniv, RA Desai, A Woehler, S Preibisch. Dual-view light-sheet imaging through a tilted glass interface using a deformable mirror. *Biomedical Optics Express*, 2021.

It would be nice if authors cited key methodological papers related to their work, including microscope development (whether they derived their design from those papers or not).

Thank you. In starting to build our microscope, we used various sources, all publicly available but not necessarily research papers. Based on this review of the existing information, we came up with many individual solutions and modifications to adjust our design according to the availability of components, our specific needs, and the technical progress. Although we cannot single out individual papers that were used as an initial blueprint or inspiration for our design, we used the opportunity to cite papers that have detailed similar setups for the readers' information.

- which XYZ stage was used for sample motion?

This is definitely an important information: We used an 8MT167-25LS XYZ 3 axis translation system from Standa (Vilnius, Lithuania) that was controlled *via* a TANGO 3 PCI-E card from Märzhäuser (Wetzlar, Germany). The information is now included in the manuscript.

- the AHF Analysentechnik is a dealer, the manufacturer of the mentioned filters is Semrock.

Thank you! We corrected the mistake.

- is the TMO (individual organoid body) segmentation done in Imaris manually, or in a automated/scripted way?

- related, is the cell/vasculature segmentation in Imaris scripted, or done manually? If manually, this is hard to claim good scalability and possibility for automation. If scripted (batch processing), the code should be published along with the manuscript, and be available for review.

Thank you for pointing out that this was not entirely clear in our manuscript. We amended the passage in the Methods accordingly.

To answer these two questions: scripted routines were used for all segmentations and vascular tracing experiments. This guaranteed high reproducibility and the possibility of automation. Imaris allows for running these scripts in batch mode. However, the scripts were not generated by writing code (which we would otherwise, of course, provide) but via the GUI of Imaris. In the first step, a general evaluation routine was developed from the range of step-by-step methods offered by Imaris. This general routine was used in all subsequent evaluations. Only thresholds to distinguish signal from background were routinely adjusted for new batches of image stacks to accommodate variations in illumination and signal strength. Both, the detailed steps and the parameters used in these scripts are displayed in Supplementary Tables 6 + 7. Everyone with access to Imaris is provided with step-by-step information to set up the very same script we used on their workstation within minutes. We are also confident that everyone using a different evaluation software is provided with enough information to understand our approach and adjust it to fit their own environment.

Reviewer #4 (Remarks to the Author):

In this revised manuscript, the authors have taken it upon themselves to argue that they were correct in the first place, and the concerns raised are either not real concerns or irrelevant to their conclusions.

The concerns I raised were aimed at giving an opportunity for the authors to make their manuscript more scholarly and balanced by acknowledging/discussing caveats and limiting the over-interpretation of their findings. However, it is disappointing that the authors are unable to see the risk their over-interpretation poses and are also unable to recognize the caveats of their studies in a scholarly manner. I am choosing not to provide a detailed response because the authors have not addressed my original concerns. I will, however, provide a couple of examples to explain myself and will leave it to the journal and authors to associate themselves with the impact (positive or negative) this manuscript will have in the field.

The authors are unable to understand the difference between cancer cells with

epithelial or mesenchymal properties and the cell type the cancer originates from. The end cell phenotype does not need to be identical to the originating cell phenotype. An epithelial cell-derived tumor may lose all its epithelial properties during its genesis and become a mesenchymal cell. So, of course, no one will question the origin, but they will know what the end phenotype is. When I indicated MDA-MB231 are mesenchymal in nature and do not have any classical epithelial properties (e.g., e-cadherin expression and cobblestone morphology), I was not referring to their originating cell type. But the authors chose to respond with a lecture on how it is a mammary adenocarcinoma, etc. etc. To me, their response highlights their dogmatism more than anything else.

The minor concern about the organoid nomenclature of what is considered in the field to be a norm was to encourage them to consider their caveats and de-risk themselves from being received by readers as scientists who do not recognize the norm for the field. In response, the authors provide a response that is largely unscholarly.

Lastly, when I highlighted most of the drugs that have failed clinical testing in breast cancer, it was to give the authors a chance to demonstrate that their TMO platform is better than the existing platform for the validation of candidate drugs either by comparing other platforms or choosing drugs that did or did not succeed in the clinic (the end goal they are trying to model in TMO) and demonstrate that drugs that work in clinic work in their TMO model and those that did not work do not work. Taking drugs that were never tested in the clinic and show efficacy or lack thereof is a drug discovery effort and not a platform validation effort.